# Help or Hinder: Protein Host Factors That Impact HIV-1 Replication

**DOI:** 10.3390/v16081281

**Published:** 2024-08-10

**Authors:** Michael Rameen Moezpoor, Mario Stevenson

**Affiliations:** 1Department of Microbiology and Immunology, University of Miami Leonard M. Miller School of Medicine, Miami, FL 33136, USA; 2Raymond F. Schinazi and Family Endowed Chair in Biomedicine; Professor of Medicine; Director, Institute of AIDS and Emerging Infectious Diseases; Department of Microbiology and Immunology, University of Miami Leonard M. Miller School of Medicine, Life Science Technology Park, 1951 NW 7th Avenue, Room 2331B, Suite 200, Miami, FL 33136, USA; mstevenson@med.miami.edu

**Keywords:** Human Immunodeficiency Virus (HIV-1), host factors, restriction factors, replication, infection, CD4+ T lymphocytes, myeloid cells, monocytes, macrophages

## Abstract

Interactions between human immunodeficiency virus type 1 (HIV-1) and the host factors or restriction factors of its target cells determine the cell’s susceptibility to, and outcome of, infection. Factors intrinsic to the cell are involved at every step of the HIV-1 replication cycle, contributing to productive infection and replication, or severely attenuating the chances of success. Furthermore, factors unique to certain cell types contribute to the differences in infection between these cell types. Understanding the involvement of these factors in HIV-1 infection is a key requirement for the development of anti-HIV-1 therapies. As the list of factors grows, and the dynamic interactions between these factors and the virus are elucidated, comprehensive and up-to-date summaries that recount the knowledge gathered after decades of research are beneficial to the field, displaying what is known so that researchers can build off the groundwork of others to investigate what is unknown. Herein, we aim to provide a review focusing on protein host factors, both well-known and relatively new, that impact HIV-1 replication in a positive or negative manner at each stage of the replication cycle, highlighting factors unique to the various HIV-1 target cell types where appropriate.

## 1. Attachment

Human Immunodeficiency Virus type 1 (HIV-1) is a positive sense, single-stranded (ss) RNA virus. Its spherical virions are comprised of two copies of the viral genome in the center, around which is the protein capsid, which itself is enclosed by a lipid bilayer envelope derived from the producer cell from which the virion budded. The HIV-1 RNA genome consists of five genes for structural and regulatory proteins, Gag, Pol, Env, Tat, and Rev; four genes for accessory proteins Nef, Vif, Vpu, and Vpr; and is flanked by 5′ and 3′ untranslated terminal regions (UTRs) [1,2,3]. From the surface of the virion envelope, the spike protein Env protrudes, composed of the surface glycoprotein gp120 connected to the viral membrane via the transmembrane protein gp41. gp120 is the predominant facilitator of interaction between the virion and the HIV-1 target cells, binding the target receptor to enable a conformational change of gp41 that initiates fusion of the viral envelope with the host cell plasma membrane (PM), releasing the viral capsid into the cytoplasm [4,5,6,7].

The host cell surface protein CD4 is the primary receptor with which gp120 interacts and is expressed on a wide variety of immune cells, such as CD4+ T lymphocytes (T cells) and myeloid lineage cells such as macrophages, microglia, and dendritic cells (DCs). CD4 is much more prevalent on CD4+ T cells than on myeloid cells. [7,8,9,10]. While CD4-independent infection has been demonstrated in vitro, such as via an endocytotic route, the infection is far less efficient than in CD4+ T cells. Interestingly, HIV-1 has been seen to bind to CD4-negative cells for cell–cell transfer into T cells for productive infection (defined as infection that results in creation of infectious progeny) [11,12,13,14]. CD4+ T cells constitute the main target for HIV-1 infection, being the most commonly infected cell type circulating in the blood, with HIV-1 preferentially infecting activated or memory T cells over naïve or quiescent T cells [15,16,17,18,19]. However, activation of T cells is not required for infection to occur [20,21]. Indeed, infection in resting T cells is a source of the long-term viral reservoirs that pose a challenge to complete viral clearance [22,23].

Myeloid cells comprise another major target for HIV-1; monocytes, macrophages (both monocyte-derived and tissue resident), microglia, and dendritic cells, have all been observed to be productively infected by HIV-1 [24,25,26,27,28,29,30,31]. As noted above, myeloid cells have less CD4 available than CD4+ T cells for HIV-1 to bind to [10,32], requiring a higher binding affinity of gp120 for CD4. This constitutes the first major barrier to HIV-1 infection of myeloid cells [33,34,35].

While the presence of CD4 is critical for the binding of HIV-1 to its target cell, the virus requires a co-receptor for the complete membrane-membrane fusion between virion and host cell [13,36]. Once gp120 binds to CD4, Env undergoes a conformational change that allows gp120 to bind with a co-receptor on the host cell surface [4,7,9]. The two main co-receptors used by HIV-1 are the surface proteins CCR5 and CXCR4 [37,38,39,40]. Those HIV-1 variants that use only CCR5 as a co-receptor are denoted as R5 viruses, those that use only CXCR4 are deemed X4 viruses, and there exists variants able to use both CCR5 and CXCR4 aptly categorized as R5X4, or “dual-tropic”, viruses [41].

CCR5 is expressed on memory CD4+ T cells, macrophages, microglia, and dendritic cells [10,27,42,43], and is the co-receptor used by “transmitted/founder” (T/F) viruses passed from one individual to another to establish initial infection [44,45,46,47]. Memory T cells express CCR5 at higher levels than naïve T cells [10,47,48,49], and thus are the predominantly infected cells during early infection. Despite the presence of CCR5 on myeloid cells, not all R5-tropic viruses are capable of infecting cells of the myeloid lineage due to the low density of CD4 on these cells. Only viruses with a high affinity for CD4 can effectively bind and infect myeloid cells. In fact, it has been found that X4 viruses can also infect both monocyte-derived macrophages (MDMs) and tissue resident macrophages despite their low levels of CXCR4, although replication is less efficient and apoptosis occurs at much higher rates when compared to M-tropic virus infection of MDMs, implying that CD4 affinity, more than co-receptor usage, determines the ability to infect different cell types [50,51,52]. Thus, those that can infect macrophages are deemed “M-tropic”, and the rest are T cell tropic (T-tropic) [53].

As infection continues, CD4+ T cell levels deplete [54,55,56]. T cell populations of long-infected individuals is predominated by naïve and resting T cells, which express high CXCR4 and low CCR5 levels [47]. This change in co-receptor abundance provides selection pressure onto HIV-1, and indeed a rise in X4 T-tropic virus is seen to correlate with chronic infection [48,57,58], as well as in vitro when CCR5-mediated binding is inhibited [59]. Memory and naïve T cells also express L-selectin (CD62L), which can bind gp120 to facilitate fusion and aid their preferential infection of these T cell subsets [60,61].

Dendritic cells express CD4, as well as CCR5 and CXCR4, allowing for virion Env binding to the DC surface, although the low level of CD4 expression limits efficient CD4 binding to M-tropic variants [27,62]. However, DCs also possess various other receptors that mediate HIV-1 binding, such as the mannose receptor (MR) and DC-SIGN, circumventing the requirement for CD4 affinity [63]. While productive infection of dendritic cells is less common and less efficient than infection of CD4+ T cells [64], DCs provide an effective transmission route to T cells via “trans-infection”, preferentially to activated CD4+ T cells [26,65].

## 2. Fusion

Fusion of the HIV-1 envelope with the plasma membrane of the host cell requires more than the initial interaction between Env and CD4/CCR5/CXCR4, and various host factors play a part in ensuring a successful fusion occurs. Intercellular Adhesion Molecule 1 (ICAM-1, or CD54) is a glycoprotein expressed on the surface of epithelial and immune cells and is found to be incorporated in the viral envelope during budding. ICAM-1 incorporation was found to increase rates of virion binding and uptake of both primary and lab-generated HIV-1 strains. ICAM-1 incorporation also reduces sensitivity to neutralization from anti-Env glycoprotein antibodies, although this did come at the cost of being sensitive to anti-ICAM-1 antibodies [66,67,68]. Adaptor Protein Complex 2 (AP-2) is a heterotetrameric protein found on the PM involved in endocytosis of the host cell’s surface receptors and plays a role in HIV-1 envelope fusion. The cytosolic domain of gp41 contains two tyrosine-based motifs, and a nearby glycine residue, that can interact with the mu-2 domain of AP-2 and facilitate internalization of the virion particle [69,70]. The binding of gp120 to target cell receptors triggers intracellular signaling that affects HIV-1 fusion. A recent study found that binding of gp120 to CD4 triggered Ca2+-dependent signaling of the lipid scramblase transmembrane protein 16F (TMEM16F), which in turn causes the externalization of membrane lipid phosphatidylserine (PS). Blocking PS externalization or inhibiting TMEM16F function resulted in severely attenuated viral fusion [71]. PS found on the surface of virions, acquired from the virion’s prior budding from an earlier infected cell, are a major factor for efficient macrophage infection. Blockage of virion-associated PS by annexin V severely attenuated infection of MDMs and monocytic cell lines but had no effect on CD4+ T cells or T cell lines. Binding of virions to MDMs/monocytic cell lines was not impacted, showing that PS does not play a role in binding, but rather in a subsequent fusion step involving a PS receptor on the cell surface [72].

Should HIV-1 effectively bind to a cell, preventing virion fusion is the next chance to avoid infection. The interferon-induced transmembrane (IFITM) protein family of interferon stimulated genes (ISGs) has demonstrated antiviral activity at multiple steps in the HIV-1 replication cycle, first and foremost by inhibiting entry. IFITM anti-entry activity is possibly due to disruption of endosomal cholesterol homeostasis, creating environments less conducive to enveloped-virion fusion and thus preventing entry [73]. IFITMs 2 and 3 have greater effects against virion entry than IFITM1, most likely attributable to their relatively extended N-terminal region [74]. However, co-receptor usage is a determinant of HIV-1 sensitivity to the different IFITMs. CCR5-using HIV-1, the transmitter/founder viruses, are more sensitive to inhibition by IFITM1 and more resistant to IFITMs 2 and 3, whereas CXCR4-utlizing HIV-1 is more sensitive to IFITMs 2 and 3 than to IFITM1. This is linked to the sub-cellular localization of each IFITM, with IFITM1 being located on the cell surface and IFITMs 2 and 3 in endosomal compartments [75,76,77]. Interestingly, IFITMs can be transferred via extracellular vesicles (EVs) from infected T cells to uninfected T cells that have yet to have their intrinsic immune response stimulated, thus supplying nearby cells with defenses for the oncoming invasion of virions [78].

Ironically, sometimes prevention of HIV-1 entry has the best chance of occurring after HIV-1 has already infected, replicated in, and budded out from, another host cell. During this process, IFITMs can be incorporated into virions, and can enact their anti-entry activities from within [79]. In the same vein, the PM-localized host restriction factor SERINC5 (and to a lesser extent, SERINC3) is incorporated into the virion, and has at least two observed antiviral effects from this position: impairment of fusion pore formation, and an increased sensitivity to neutralizing antibodies. These effects vary in severity depending on the HIV-1 isolate, as the Envs of some strains show enhanced resistance to SERINC5 [80,81,82,83]. Similar to SERINC proteins, P-selectin glycoprotein ligand-1 (PSGL-1) is a host protein highly active during inflammation normally found on leukocytes to enable attachment onto, and migration into, endothelial tissues. PSGL-1 acts as a restriction factor after being packaged into HIV-1 virions, as it then inhibits the ability of the virion to infect the next target cell. Its inhibitory effects stem from its extracellular N-terminal region, as deletion of this domain removes the block against virion entry [84,85]. As implied by the antiviral activity of IFITMs, cholesterol homeostasis is a key element of successful fusion. The cellular cholesterol transporter ATP binding cassette transporter A1 (ABCA1), also known as cholesterol efflux regulatory protein (CERP), controls the efflux of cholesterol from cells. This is to the detriment of HIV-1, as lower cellular cholesterol levels reduce cholesterol incorporation into budding virions, which has been demonstrated to negatively impact infection at the stage of fusion. Nef-deficient HIV-1 strains are particularly vulnerable to ABCA1′s effects, as Nef suppresses ABCA1 function in CD4+ T cells and macrophages by inducing its lysosomal degradation [86,87,88,89].

## 3. Capsid Transport and Uncoating

After membrane fusion, the conical viral capsid is released into the cytosol where it must gain entry to the nucleus. Transport of the HIV-1 genome across an intact nuclear membrane is a key factor in productive infection of non-dividing cells, such as resting CD4+ T cells and macrophages, removing the requirement for nuclear dissolution during cell division that other retroviruses may require [90]. While past models of HIV-1 replication involved the capsid uncoating in the cytosol or at the nuclear pore [91,92,93], there is mounting evidence that the capsid is transported to the nucleus and enters before completely uncoating [94,95,96,97,98]. Beyond facilitating genome transportation, the capsid shields the viral genome from host restriction factors and sensors of foreign DNA that would otherwise raise the alarm of the innate immune system [99,100].

A wide variety of host factors facilitate successful capsid transport, import into the nucleus, and uncoating for viral genome release (Figure 1). Sufficient capsid stability ensures not only proper interactions with these host factors but prevent recognition of the viral genome by host foreign DNA sensors. Sec24c and PDZD8 bind the outside of the capsid to enhance capsid stability [101,102]. Cyclophilin A (CypA) is a well-characterized host factor with peptidyl prolyl *cis*-*trans* isomerase activity that regulates protein folding and trafficking, and is found intracellularly but can be secreted in response to inflammatory signals [103]. CypA is packaged into the virion during viral assembly, binding the HIV-1 Gag polyprotein via the capsid protein (CA), and has been shown to be a crucial host factor for infectivity [104,105,106,107]. Researchers report that CypA promotes capsid stabilization during transport across the cell [108,109,110,111,112,113] and may provide resistance against DNA sensors and antiviral factors such as TRIM5α [114,115]. Some researchers suggest that CypA is involved in the destabilization of the capsid for proper genome release, possibly in a cell type-specific manner [110,116,117]. In either case, CypA is shown to be beneficial to HIV-1 replication, as virions lacking the protein lead to decreased infection productivity. Inositol hexakisphosphate (IP6), though not a protein, was also recently characterized as a host factor packaged into the HIV-1 virion, promoting capsid stability and preventing recognition by DNA sensors [118,119]. BICD2 and FEZ1, adaptor proteins for the microtubule (MT)-associated proteins dynein and kinesin-1, respectively, bind to the capsid to expedite cytoplasmic trafficking via MTs [120,121,122,123]. It should be noted that while interactions with MTs have been established as an important method of capsid transport to the nucleus, MT-independent trafficking is still a viable route for HIV-1 [124]. Nup358 and Nup153, components of Nuclear Pore Complexes (NPCs) mediate nuclear import [125,126]. Cleavage and polyadenylation specificity factor 6 (CPSF6) is another crucial CA-binding host factor, as it regulates nuclear entry through NPCs, as well as localization of the pre-integration complex (PIC, discussed later) to open chromatin, the sites of active transcription, ensuring successful generation of vial mRNA [127,128,129,130,131,132,133]. Once in the nucleus, nuclear-localized proteins Transportin-1 (TNPO1), transportin-3 (TNPO3), and Pin1, trigger capsid destabilization and complete uncoating [108,134,135].

Multiple host restriction factors work to impede HIV-1 capsid functions (Figure 1). Tripartite motif-containing protein 5α (TRIM5α) is an E3-ubiquitin ligase and ISG that promotes premature virion disassembly, preventing reverse transcription from completing and allowing recognition and destruction of the unprotected viral genome. However, TRIM5α’s antiviral properties function in a species-specific manner. TRIM5α of old world monkeys was found to be effective against HIV-1, being able to bind the capsid and cause uncoating, whereas human TRIM5α is unable to effectively bind the HIV-1 capsid [136,137,138,139,140]. Exceptions to TRIM5α’s species-specific antiviral activity include virions that insufficiently package CypA prior to infection [115], and TRIM5α-mediated degradation via autophagy in Langerhans cells (a subset of dendritic cells located in mucosal epithelia of genitals and considered a barrier against pathogens) [141]. TRIM5α also plays a role in the immune sensing of HIV-1 by forming a “TRIM5 cage” that leads to auto-ubiquitination, promoting viral capsid detection and stimulation of the innate immune response, as well as proteasomal-mediated degradation [142,143,144]. Likewise, cGAS adaptor protein PQBP1 was recently described to enhance immune detection by binding the capsid, facilitating viral genome detection by cGAS once the genome is released [145,146], and another TRIM family protein, TRIM11, causes premature capsid uncoating [147]. MX2, also known as MXB, is an ISG found both within and around the nucleus that binds the virion capsid and prevents nuclear import. MX2 may also inhibit uncoating, preventing viral genome release [148,149,150,151,152,153]. The restriction factor Daxx, found mainly in the nucleus, also inhibits capsid uncoating by interacting with both CypA and CA, in a SUMOylation-dependent manner [154].

## 4. Reverse Transcription

During the capsid’s trek from the membrane to the nucleus, the viral genome must be reverse transcribed by RT from positive-sense ssRNA to double-stranded (ds) DNA, with the 5′ and 3′ UTRs being reverse transcribed into 5′ and 3′ long-terminal repeats (LTRs) [3]. The process of reverse transcription, particularly an intermediate step involving first-strand transfer, is shown to be critical for capsid uncoating to occur, as inhibition of RT produced extra-stable capsids that could not disassemble and release their payloads [91,155,156]. The space within the capsid is known as the viral core, and packaged along with the vial genome are the viral RT, nucleocapsid protein (NC), integrase (IN), and Viral Protein R (Vpr). Post-fusion, some matrix protein (MA) is also associated with the capsid, although whether or not it is incorporated inside the core is unknown. Together these components comprise the reverse transcription complex (RTC) of HIV-1 [157]. Cellular proteins that benefit HIV-1 like CypA, and restriction factors like APOBEC3G (discussed below) are also found within the capsid [158]. Pores in the capsid allow deoxynucleotide triphosphates (dNTPs) to enter and provide the building blocks RT uses to create viral cDNA, utilizing the viral RNA genome as a template [159]. Reverse transcription is carried out almost entirely by the DNA polymerase and RNAse H enzymatic activities of RT, with NC being involved in a strand-transfer step in the process [160].

The host cell cytoskeleton, particularly the actin microfilament network, plays an important role in initiating reverse transcription. MA interacts with actin soon after entry to localize the RTC to the microfilaments, after which efficient reverse transcription occurs. Disruption of actin or the MA-actin interactions directly impacts the abundance of newly generated viral DNA [161]. Many host factors that ensure efficient reverse transcription are often packaged into the virion to assist in the process prior to uncoating. Histone deacetylase 1 (HDAC1) is packaged into HIV-1 virions by virtue of the complex it forms with another host factor, sin3A-associated protein 18 (SAP18). SAP18′s interacts with IN, thus leading to the SAP18-HDAC1 complex’s virion packaging. HDAC1 promotes optimal levels of reverse transcription, as knockdown or silencing of HDAC1 results in decreased amounts of reverse transcripts [162,163]. As its name suggests, integrase-interactor 1 (INI1) binds IN and is implicated in efficient RT. When infected with HIV-1, the *INI1*-deficient rhabdoid cell line MON had reduced virion production, with the virions that were produced displaying reduced infectivity. This debilitation was tracked to the reverse transcription step, as RT productivity was decreased in cells infected with INI1-deficient virions compared to those containing INI1 [164]. Topoisomerase I (TOP1) is a packaged host factor that breaks and rejoins single strands of DNA, relieving tension. TOP1 interacts with the NC of the RTC to ensure efficient RT activity, with inhibition of TOP1 by camptothecin showing a marked reduction in viral cDNA [165]. The RNA helicase Up-frameshift protein 1 (UPF1), ubiquitously found in cells as part of the mRNA Nonsense-Mediated Decay (NMD) pathway, is also packaged in virions and is involved in maximizing the quantity of reverse transcripts produced, a quality linked to its ATPase activity [166,167].

The mRNA-stabilizing ELAV-like protein HuR, while not observed to be packaged into HIV-1 virions, was shown to positively modulate RT activity by protein-protein interaction with RT’s RNase H domain. Silencing of HuR resulted in impaired reverse transcription [168]. Similarly, A-kinase anchoring protein 149 (AKAP149, AKAP1), which normally associates with protein kinase A (PKA) to spatially regulate its activity, also interacts with RT via its RNase H region. Silencing of AKAP149 impairs HIV-1 replication at the reverse transcription step; however, AKAP149′s exact role in the process is not yet known, though it may relate to its usual associations with PKA, which has also shown to impact HIV-1 replication [169,170]. Survival motor neuron (SMN)-interacting protein 1 (Gemin2) is a mediator of spliceosomal small nuclear ribonucleoprotein (snRNP) assembly and binds to IN to positively regulate reverse transcription in MDMs, most likely by promoting reassembly of the RTC for continuous production of viral cDNA [171,172]. Interactions between these factors and viral proteins that would theoretically be isolated within capsid lends credence to the possibility that while capsid uncoating may occur in full within the nucleus, as recent evidence suggests (see Section 3), it may begin to some degree in the cytoplasm and allow host factor access to these internal proteins.

Reverse transcription’s role as a trigger for capsid uncoating also provides more targets of opportunity for the host cell to exploit. Without the capsid to act as a bulwark, the threat of viral RNA or DNA detection by cellular sensors becomes a major factor in the viral replication cycle. TREX1 is an exonuclease that degrades HIV-1 DNA, thus preventing immune sensing of the newly made viral genome [173,174,175]. ADAR1 is an RNA-editing ISG that was found to have pro-virus effects. ADAR1 can edit viral RNA by deaminating adenosine and thus hide it from sensors like MDA5 and RIG-I that would otherwise detect it, and also function as an inhibitor of PKR, an antiviral ISG (discussed below) [176,177,178]. Once the transition to dsDNA is completed, the association of genome and proteins is dubbed the pre-integration complex (PIC). Host factor ADAM10 possesses a nuclear localization signal and interacts with the IN of the PIC and is thought to facilitate shuttling of the PIC to and possibly through the nuclear pore complex, in macrophages and quiescent CD4+ T lymphocytes. This supports the original notion that, to some degree, capsid uncoating occurs, or at least begins, prior to breaching the nucleus [98,179,180,181]. IN is also seen to interact with importin α3 (karyopherin α4) to facilitate PIC nuclear localization and importation [182,183].

Various host restriction factors work to interfere with the creation of viral cDNA and PIC activities. In CD4+ T cells and monocytes/macrophages, the sensing of viral dsRNA is facilitated by RIG-I (with some evidence possibly implicating MDA5), which signals through MAVS to induce an interferon response [184,185,186,187]. Monocytes and macrophages also show viral sensing of ssRNA via toll-like receptor (TLR) 8 [188]. IFI16 and cGAS recognize vial dsDNA and activate the stimulator of interferon genes (STING) pathway, although there is some debate over cGAS’s role [189,190,191,192,193]. While myeloid dendritic cells can use cGAS to detect HIV-1, this is dependent on capsid-CypA interactions [194,195]. Plasmacytoid dendritic cells can sense HIV-1 RNA through TLR7 in an endocytosis-dependent manner [196]. Regardless of the RNA/DNA sensing pathway, the end result is the expression and release of interferons that in turn promote the expression of antiviral ISGs that counter viral infection in infected cells. Secreted interferons can also bind to interferon receptors of neighboring uninfected cells, priming them for the oncoming virus and inducing expression of ISGs prior to infection. In addition, some ISGs, such as IFITM and ISG15, can be released from the infected cell to uninfected cells, directly acting as defenses to the incoming virus [197,198,199]. Proteins of the apolipoprotein B mRNA editing catalytic polypeptide-like 3 (APOBEC3) family of ISGs function as cytidine deaminases, converting cytosines (C) to uracils (U), and acting upon viral and host DNA and RNA alike. Though normally considered RNA editors, they can also function as DNA mutators: when enacted upon the minus strand of the newly reverse transcribed viral cDNA, C to U conversions can result in guanine (G) to adenine (A) hypermutations on the plus strand, leading to premature STOP codons or missense mutations. Although most members of the APOBEC3 family have anti-HIV-1 abilities to varying degrees, the most prominent is APOBEC3G. APOBEC3G is unique for its preference of C’s in CC motifs, whereas the other APOBEC proteins prefer TC motifs. This preference results in more occurrences of TGG (tryptophan) to TAG (STOP) mutations. Binding of APOBEC3 proteins to viral RNA also leads to their packaging into virions during assembly, where they carry out deleterious editing of HIV-1 genomic RNA post-budding [200,201,202,203,204,205,206,207,208,209,210,211,212,213,214,215]. Some reports show APOBEC3 antiviral activities are not solely linked to their mutagenic capabilities, and may in fact be attributed to directly binding to RT and acting as a roadblock to reverse transcription [216,217,218,219,220]. PSGL-1 is also an ISG that, outside of its anti-fusion effects, is postulated to negatively impact HIV-1 at a pre-integration step. PSGL-1 was observed binding to F-actin and preventing its depolymerization, thus disrupting the reverse transcription process. However, others have found PSGL-1 to have no impact at this step [85,221,222]. While not an ISG, the PAF1 complex was identified as an antiviral factor that represses HIV-1 replication prior to integration, most likely during the reverse transcription step [223]. Exonuclease 1 (Exo1), a post-replication DNA repair protein that degrades DNA in a 5′ to 3′ manner, serves as a restriction factor by degrading vial DNA [224]. Helicase-like transcription factor (HLTF) is part of DNA repair pathways in macrophages and dividing T cells. HLTF has been shown to bind the 3′ end of viral ssDNA, and most likely acts in concert with proteins like Exo1 to effectively destroy viral DNA [225]. The helicase MOV10 is in the same protein family as UPF1 but has the opposite effect on HIV-1, as it inhibits reverse transcription, and while it can be incorporated into virions this is not required for its antiviral activity [226]. MX2, along with its anti-uncoating capabilities, has been found to impede NUP358-mediated entry of the PIC when the PIC is concurrently interacting with CPSF6 [227]. REAF/RPRD2 is a restriction factor found in dendritic cells and M1/M2 polarized macrophages, and binds reverse transcripts to impede the reverse transcription process [228,229].

Sterile alpha motif and HD-domain containing protein 1 (SAMHD1) can be considered the most crucial post-entry restriction factor against HIV-1 replication in non-dividing cells. It was found that HIV-1 replicated slower in myeloid cells than in activated T cells, and that this retarded growth could be attributed to lower levels of deoxyribonucleoside triphosphates (dNTPs) available for HIV-1 to use for reverse transcription of its RNA genome into DNA [230,231,232]. Interestingly, while HIV-1 was impaired in these nondividing cells, HIV-2 and SIV that expressed viral protein X (Vpx) were not impeded, implying Vpx countered whatever would otherwise block replication. Co-immunoprecipitation experiments for proteins that interacted with Vpx identified SAMHD1, a dNTP triphosphohydrolase that controls dNTP levels when cells are not preparing for division, as the host restriction factor. Subsequent experiments with SAMHD1-deficient non-dividing cells confirmed its importance in HIV-1 restriction [233,234,235,236,237,238]. Later, the observed slower HIV-1 replication in resting T cells was also attributed in large part to SAMHD1 activity [239,240]. Due to the low intracellular dNTP concentrations, possessing an RT with the ability to efficiently reverse transcribe while in a dNTP-low environment is a necessity for HIV-1 replication in these non-dividing cells, as HIV-1 has no Vpx with which to counter SAMHD1 and has not been shown to alter SAMHD1 function in any capacity. Indeed, possession of an extremely efficient RT can be considered the hallmark of M-tropic viruses [236,241,242,243,244]. There is some debate as to the exact nature of SAMHD1′s antiviral properties. Although originally attributed to its function as a dNTPase, experiments involving phosphorylation of SAMHD1 saw a reduction in its antiviral capabilities while no change was seen in dNTP levels [245]. SAMHD1 was also found to have exonuclease activity, acting as an RNAse in the presence of DNA:RNA duplexes such as HIV-1 would produce during reverse transcription [246,247]. Thus, SAMHD1′s antiviral properties may not be attributed solely, if at all, to its depletion of intracellular dNTP levels. Regulation of SAMHD1 in both myeloid and lymphoid cells is controlled by p21, a controller of cell-cycle progression and inhibitor of cyclin-dependent kinases, which itself is a downstream gene of the tumor suppressor gene p53 [248,249,250,251]. p21 is also involved in SAMHD1-independent HIV-1 restriction in macrophages, as Fcɣ receptor induction of p21 leads to p21-mediated downregulation of the RNR2 subunit of ribonucleotide reductase, an enzyme involved in dNTP biosynthesis, further limiting the pool of available dNTPs for HIV-1 to use [252,253]. Interestingly, an indirect antiviral effect may be attributed to SAMHD1′s dNTPase activities and p21′s disruption of dNTP biosynthesis. The low dNTP levels cause a low dNTP to ribonucleoside triphosphate (rNTP) ratio in the cell, increasing the chance of incorporation of ribonucleotides into the reverse-transcribed viral genome. Not only does this cause delays in reverse transcription, but the reduced RNAse H2 and Fen1 DNA repair capabilities of macrophages result in incomplete repair of these incorporations, the end result being potentially deleterious mutations in the viral genome [254].

## 5. Integration

Once the PIC has successfully entered the nucleus, the next step in the HIV-1 replication cycle is the integration of the viral genome into the host genome [255]. It should be noted that a majority of the newly reverse-transcribed genomic DNA will not be integrated, and will continue to exist outside of the host genome. Unintegrated viral DNA can take on a myriad of forms, such as linear DNA, or circularized in 1-LTR or 2-LTR circles. These forms contribute to gene expression, interact with beneficial or detrimental host factors, and can even play a role in “preintegration latency” (reviewed in [256]).

The IN protein of the HIV-1 PIC interacts with a variety of host proteins to ensure successful integration into sites of active transcription found in uncondensed regions of chromatin known as euchromatin. Acetylation of IN greatly enhances its DNA-binding capabilities and enzymatic activities. p300 is a histone acetyl transferase (HAT) that acetylates IN’s C-terminal domain (CTD), increasing its affinity for DNA and capacity for integration [257]. As mentioned earlier, CPSF6, in addition to its role in getting the capsid and PIC to and through the nucleus, is a key player in the intranuclear localization of HIV-1 to euchromatin. Indeed, CPSF6 has been implicated in the depth of nuclear penetration of HIV-1 to the gene-rich locations, with CPSF6-KO cells showing reduced provirus formation and decreased levels of integration into sites of active transcription, reducing overall viral infectivity [128,130,133,258]. LEDGF/p75 is another factor involved in HIV-1 integration, previously considered to be the factor that trafficked HIV-1 to sites of active transcription, rather than CPSF6 [259]. However, while LEDGF/p75 was shown to not be responsible for the act of transporting HIV-1 to euchromatin, it plays a very key part in preferential integration into specific parts of the active gene bodies to which CPSF6 delivers HIV-1. LEDGF/p75 targets HIV-1 to the transcription units of specific genes, avoiding integration into promoters and CpG islands. Concurrent binding of LEDGF/p75 to IN and the euchromatin promotes integration via tethering, with LEDGF/p75-defective mutant models displaying reduced levels of HIV-1 integration into these active gene bodies, and reduced levels of HIV-1 replication [133,255,258,260,261,262,263,264,265,266]. HMG I(Y), of the High Mobility Group architectural protein family, co-fractionates with the PIC and helps HIV-1 integrate into the genome [267,268]. Interactions between IN and the SWI/SNF chromatin remodeling complex allow integration into stable nucleosomes that would otherwise be resistant, in vitro [269]. DNA-PK and its DNA-binding component Ku80, a DNA-damage response complex specific for double-strand breaks (DSBs), have been implicated in HIV-1 integration. Deletion of DNA-PK leads to abortive-integration-triggered apoptosis and under-expression of Ku80 results in impaired proviral formation [270,271,272,273]. Similarly, ATM, another protein associated with DSB DNA-damage response, promotes stable integration, with ATM knockdown mutants undergoing cell death as a result of inefficient integration [274].

The prevention of integration is the cell’s last chance to prevent HIV-1 from gaining a foothold in the cell and potentially becoming a reservoir for future reactivation, as this is an irreversible process [275]. Interestingly, INI1, reported to assist in the reverse transcription stage of HIV-1 replication, seems to stall HIV-1 at the integration step in a manner not require its prior incorporation into virions. Depletion of INI1 saw an increase in unintegrated 2-LTR circles and integrated HIV-1 DNA, leading to speculation that INI1′s interaction with IN may destabilize the PIC and interrupt IN’s activity [276]. While some DNA repair proteins aid in integration, others are involved in its prevention. RAD52, a DSB repair protein that utilizes the homologous recombination route of repair, binds the LTR portion of HIV-1 cDNA and prevents stable integration. This is done in competition with Ku80, as both these proteins will bind the HIV-1 LTR in a mutually exclusive manner [277]. TFIIH is a protein complex involved in nucleotide exclusion repair, and two of its components, the helicases XPB and XPD, have been implicated in the reduction of HIV-1 integration events. Mutant XPB or XPD cell lines showed higher cDNA kinetics than wild-type, and their antiviral activity was deduced to be due to degradation of viral cDNA, rather than interruption of cDNA synthesis [278]. Outside of DNA repair pathway proteins, KAP1 is a TRIM family protein that, similarly to SAP18, complexes with HDAC1. However, this HDAC1 interaction is to the detriment of HIV-1, as KAP1 associates with acetylated IN, promoting HDAC1-mediated deacetylation of IN, disrupting IN’s function and impairing integration [279].

## 6. Transcription

Whether integrated or existing in the nucleus as unintegrated genomic DNA, HIV-1 has three possible levels of transcription: basal expression, increased expression via the HIV-1 trans-activator protein Tat (which binds the Transactivation Response element (TAR) of nascent RNA transcripts to promote transcriptional elongation [280]), and repressed transcription to maintain a latent state. The 5′ LTR of HIV-1 contains the core promoter (a TATA element, three GC-rich Sp1 binding sites, and an initiator site), upstream elements, enhancer elements, and 5′ UTR/downstream region [281]. Many host factors are involved in modulating expression levels of HIV-1 (comprehensively reviewed in [281,282]). Transcription requires RNA polymerase II (RNAPII) to associate with the viral promoter. This is facilitated by the TATA-binding protein (TBP) portion of the TFIID transcription factor, which binds the HIV-1 TATA element to begin the formation of the preinitiation complex that positions RNAPII to begin transcription [281,283,284].

A multitude of host factors, particularly cyclin-dependent kinases (CDKs), work to ensure proper RNAPII function and mRNA processing [285]. Cdk9, part of the positive transcription elongation factor (P-TEFb) and Tat-associated kinase (TAK), can autophosphorylate and thereby activate itself, subsequently phosphorylating serine 2 of RNAPII’s C-terminal Domain (CTD) and facilitating Tat transactivation, promoting transcriptional elongation [286,287,288,289]. Host transcription factor ZASC1 was found to have a DNA binding element upstream of the TAR, recruiting both Tat and P-TEFb in a TAR-independent manner to promote HIV-1 gene expression and transcriptional elongation [290]. The expression of TAK’s components, Cdk9 and cyclin T1, is upregulated in response to T cell activation and monocyte differentiation into macrophages, directly impacting HIV-1 transcription levels [291]. TFIIH has also been shown to associate with the HIV-1 preinitiation complex, through interaction with Tat (although its interactions are not exclusively Tat-mediated), and is implicated in Tat-mediated transcriptional elongation, although whether it is necessary has not been resolved. In preinitiation complexes lacking Tat, the Cdk7 subunit of TFIIH will phosphorylate serine 5 of RNAPII’s CTD; in the presence of Tat, Cdk9 undergoes Tat-induced autophosphorylation and activation, and its substrate specificity is altered so that it phosphorylates both serine 2 and serine 5 of RNAPII’s CTD, resulting in prolonged CTD phosphorylation even after TFIIH is released from the complex. While TFIIH is still associated with the preinitiation complex, Cdk9′s autophosphorylation is forestalled by the XPB subunit of TFIIH, revealing dynamic interactions between the factors involved in HIV-1 transactivation [292,293,294,295,296]. The RNAPII transcriptional co-activator p15 (PC4), the nuclear DNA-binding protein NUCKS1, and the TAR RNA binding protein (TRBP) are also Tat-associating host factors that play a role in promoting Tat-mediated transcription [297,298,299,300].

Both capping and splicing of viral mRNAs require the aid of host factors. Phosphorylation of RNAPII’s CTD’s serine 5 by Cdk9 also causes the recruitment of mRNA capping enzyme (Mce1) and subsequent co-transcriptional 5′ capping of the viral mRNA. Mce1 can also be recruited directly by Tat, although this capping pathway is lower efficiency than the conventional RNAPII-mediated route [301,302,303,304,305]. Alternative splicing of the HIV-1 provirus results in the production of over 50 mRNA isoforms. An assortment of splicing regulatory elements (SREs), such as HNRNPF and SRSF2, are responsible for generating these isoforms (reviewed in [306]). When splicing is not desired, the hnRNP A/B proteins are used to inhibit mRNA splicing [307,308]. Recently, it was found that protein kinase RNA-activated (PKR), which is an antiviral RNA immune sensor, can be co-opted to splice HIV-1 rev/tat mRNA. Binding the mRNA activates PKR and induces mRNA splicing, as well as PKR’s antiviral activities (discussed below) [309].

The transcription factor specificity protein 1 (Sp1) binds to the Sp1 binding sites and is needed for basal transcription rates, with HIV-1 transcriptional efficiency showing marked reductions when Sp1 binding sites are mutated, as well as for Tat-activated levels of transcription [310,311]. Other members of the Sp family can also utilize these Sp1 binding sites: like Sp1, Sp4 acts as an activator of the LTR whereas SP3 is a repressor [312]. Non-Sp proteins, such as the transcription factor BTEB, can also bind the Sp1 binding sites and activate the LTR [313].

The enhancer site upstream of the Sp1 binding sites is composed of two NF-кB binding sites, wherein NF-кB binds to augment HIV-1 transcription. NF-кB-mediated enhancement occurs in both activated T cells and macrophages. Interestingly, infected promonocytes (such as the U937 cell line) only show NF-кB binding once they begin maturing to monocytes [314,315]. Though not required for NF-кB to bind the enhancer element, interactions between NF-кB and Tat promote greater Tat-mediated transactivation of transcription. In fact, Tat can indirectly induce the expression of NF-кB to start a perpetual positive feedback loop of enhanced viral transcription [316,317,318]. As the term “enhancer” implies, NF-кB amplifies the rates of transcription but is not strictly required for HIV-1 to successfully transcribe and replicate [314,319,320,321]. NF-кB -mediated enhancement of HIV-1 transcription requires binding to the TATA element via TBP and TFIIB, as well as to Sp1. Sp1 binding is specific, as neither Sp3 nor Sp4 can interact with NF-кB [312,322,323]. Cytokines such as TNFα, IL1, TGF-β, and IL-10, and phorbol esters like TPA, can all stimulate the enhancer by induction of nuclear factor binding to the NF-кB binding sites [324,325,326,327,328,329]. Binding to the enhancer element is not limited to just NF-кB, nor does it always result in enhancement of transcription. Transcription factor Ets can interact with NF-кB while binding the enhancer to stimulate transcription, while cell cycle coordinator EF2-1 prevents NF-кB binding to downregulate enhancer-mediated, and even basal level, transcription [330,331].

Binding sites for factors affecting transcription can be located not just in the 5′LTR of HIV-1, but intragenically as well. An intragenic enhancer element was located in the *pol* gene of HIV-1, and within this element are three binding sites for AP-1 transcription factors (JunD, JunB, c-Fos) that positively regulate HIV-1 transcription. Mutations in these sites lead to a marked decrease in HIV-1 replication in promonocytic and T cell lines, as well as MDMs. This decrease was attributed to reduced recruitment of RNAPII to the viral promoter [332,333,334].

Although HIV-1 transcription in myeloid cells and T lymphocytes use many of the same host factors, cell-type-specific transcription factors play a role in controlling HIV-1 transcription. NFAT-1, a T cell-specific factor, binds the upstream region of the LTR to negatively regulate transcription. NFAT has also been seen binding to a downstream site in the 5′UTR to induce positive effects on transcription [335,336]. GATA3 is a factor essential for T cell development, and LEF-1 activates the T-cell receptor; both have been shown to have binding sites in the HIV-1 LTR to enhance transcription [337,338]. Binding sites for CCAAT/enhancer binding proteins (C/EBPs) are found in the HIV-1 5′ LTR. Intact C/EBP sites and the proteins that bind them are essential for both basal and enhanced transcription levels in monocytes and macrophages [339,340,341].

Repression of HIV-1 transcription can be both beneficial and detrimental; on the one hand, it prevents HIV-1 replication and spread of infection, while on the other it allows latent HIV-1 to stay hidden and persist for years until reactivation. Many host factors mediate the repression of HIV-1 transcription. CTIP2 complexes with P-TEFb to act as a negative regulator of P-TEFb activity, thus impairing the elongation of HIV-1 RNA during transcription [342,343]. p53, in addition to its regulation of p21 and ultimately SAMHD1, inhibits transcription by interacting with proteins that associate with the Sp1 and TATA elements of the core promoter, although this inhibition is mitigated when the NF-кB sites are functional [344]. LEDGF/p75, past its role for specific integration of the HIV-1 genome into active gene bodies, is recruited by Spt6 to the HIV-1 promoter to form a complex with Iws1 and silence the HIV-1 provirus and prevent reactivation [345]. Oct1 and Oct2 are factors that can bind to multiple octamer sites in the LTR, repressing basal and Tat-mediated activation of the LTR [346]. Uracil DNA glycosylase 2 (UNG2), a DNA repair protein that excises uracil bases from DNA, negatively regulates LTR transcription when stimulated by Tat [347]. The transcription factor YY1 binds a site near the HIV-1 initiator sequence to downregulate transcription [348]. LBP-1 (LSF) is noteworthy as it has two binding sites in the LTR, one downstream of the promoter and one overlapping the TATA element, with opposing effects on HIV-1 transcription. Binding to the downstream site is implicated in basal promoter activity, whereas binding at the TATA-overlapping site (unless outcompeted for the site by TFIID) repressed the elongation step of transcription. LBP-1 repression of transcription also seems to require interaction with YY1. Both LBP-1 binding events happen simultaneously in an LBP-1 concentration-dependent manner, suggesting a balancing act for controlling HIV-1 transcription [349,350,351,352]. IFI16, previously described here as an immune sensor against HIV-1, plays a direct antiviral role by binding Sp1 to prevent its binding and activation of the LTR, which also suppresses reactivation of latent HIV-1 [353]. TRIM22 similarly prevents transcription, and reactivation from latency, by inhibiting binding of Sp1 to its binding sites [354,355,356]. In addition to its anti-fusion activity, SERINC5 has also been observed to downregulate HIV-1 transcription, although the molecular mechanisms behind it are still unknown [357]. The SMC5-SMC6 complex localization factor 2 (SLF2) specifically targets unintegrated DNA. SLF2 recruits the SMC5/6 complex to unintegrated HIV-1 DNA, in any of its forms, and physically compacts the viral chromatin to decrease its accessibility and reduce its activating histone markers, thereby silencing gene expression by inducing a heterochromatin-like state onto unintegrated HIV-1 DNA [358].

The inhibition of transcription can also be caused by physical barriers. Chromatin reassembly factors (CRFs) can restructure HIV-1-integrated DNA to occlude polymerase activity, thereby ensuring a latent state, with depletion of these factors leading to reactivation [359]. HDACs remove acetyl groups on the lysines of histone tails, causing chromatin remodeling resulting in obstruction of access to associated DNA. This inhibits transcription of HIV-1, and HDACs have long been considered monumental contributors to the latency of HIV-1 [360]. In addition to associating with P-TEFb to repress transcription, CTIP2 is part of a chromatin remodeling complex, and through interactions with lysine methyltransferases and HDACs can restructure the architecture of chromatin to silence HIV-1 expression in microglial cells, contributing to HIV-1 reservoirs in the brain [361]. ZBTB2 is recruited to the nucleus by ZASC1, the latter of which was mentioned earlier for its recruitment of Tat and P-TEFb to an element upstream of TAR. ZBTB2 represses HIV-1 gene expression through interactions with HDACs to cause deacetylation of histones, leading to silencing of HIV-1 genes. Interestingly, ZBTB2 is no longer recruited to the nucleus or is removed from the HIV-1 promoter if already associated, if the ATR DNA damage response pathway is stimulated [362].

Should inhibition of transcription fail, the degradation of viral mRNA serves as a different means to the same end. RNAse L acts to control RNA levels, degrading viral and cellular RNAs alike. HIV-1 infection induces RNAse L Inhibitor (RLI, or ABCE1) which helps to counter RNase L-mediated transcript destruction [363,364,365,366]. In CD4+ T cells, the long isoform of ZAP (ZAP-L) is an ISG that, when complexed with TRIM25 and KHNYN, exerts antiviral activity by binding to high-CpG portions of multiply spliced HIV-1 mRNA and recruiting other host factors (poly(A)-specific ribonuclease, RNA exosome, and RNA helicase p72) to facilitate mRNA degradation. It should be noted that only CpG-enriched variants of HIV-1 appear to be significantly affected by ZAP [366,367,368,369,370,371,372,373,374]. N4BP1 is an ISG of both macrophages and CD4+ T cells that degrades viral mRNA [375]. MCPIP1 (otherwise known as regnase-1 or Zc3h12a) is an RNAse that degrades viral mRNAs in quiescent T cells [366,376]. Dicer and components of RNA-induced silencing complexes (RISCs), such as Ago1 and Ago2, are all involved in the creation of microRNAs that target viral mRNA for destruction. Interestingly, high expression of microRNA alone may be a key determinant of monocyte and macrophage susceptibility to HIV-1 infection [377,378,379,380].

## 7. Nuclear Export of Viral mRNA

Like all mRNA, the successfully transcribed viral RNA must be exported from the nucleus HIV-1 worked so hard to enter before it can be translated into proteins by the cell’s ribosomes. HIV-1 mRNA contains splicing sites, with fully spliced mRNA being exported via the conventional binding by NXF1 and the EJC for transport to the cytoplasm [380,381]. For unspliced and partially spliced mRNA this export route is unavailable. However, one of the fully spliced mRNAs is for the viral protein Rev, which will be exported, translated, and imported into the nucleus via its NLS, facilitating a shift from highly spliced to less spliced mRNA. Rev’s NLS interacts with a variety of host factors to facilitate its nuclear localization, such as importins beta, 5, and 7, as well as the transport receptor transportin [382].

Once back in the nucleus, Rev binds to the Rev Response Element (RRE) of unspliced/partially spliced viral mRNA and contains a leucine-rich Nuclear Export Signal (NES). This NES is where host factor CRM1 (also known as exportin-1 or XPO1) a nuclear export receptor, binds to Rev in a co-transcriptional manner. Binding to the 5′ cap of the mRNA is the nuclear-cap-binding complex (CBC), an export complex that associates with CRM1 and unspliced small nuclear RNA (snRNA) to mediate its transport to the cytoplasm, as well as later steps in translation (discussed further below) [383,384,385,386,387]. With Rev bound to the RRE of the unspliced/partially spliced viral mRNA, and with the aid of RanGTP to stabilize the interaction between CRM1 and cargo, the Rev-CRM1-mRNA complex is exported out of the nucleus [388,389,390,391,392,393,394,395]. The proteins PACS1, PSF, MATR3, RAB, RNA helicase A, RBM14, DDX3, DDX1, Sam68, eIF4A, hnRNP A1, and hRIP have all been implicated in the Rev-CRM1 route of nuclear export, as depletion of any of these proteins decreases the quantity of unspliced HIV-1 RNA detectable in the cytoplasm [396,397,398,399,400,401,402,403,404,405,406,407].

Interestingly, unspliced HIV-1 mRNA also recruits various host proteins usually involved in mRNA decay pathways to aid in mRNA export. UPF1, mentioned above as a positive factor in reverse transcription, is involved in NMD of prematurely truncated mRNA and is recruited by HIV-1 post-transcription to shuttle unspliced mRNA safely from the nucleus to the cytoplasm [380,408,409]. STAU1 and STAU2 are proteins of the Stau-Mediated Decay (SMD) pathway of mRNA degradation that interact with Gag and Rev, respectively, as well as the unspliced mRNA to guarantee nuclear export and translation [380,410,411,412,413].

## 8. Translation

Translation of the HIV-1 mRNA into proteins is the next step of the actively replicating virus’s life cycle. As with all protein synthesis, this requires effective and accurate recruitment of the host ribosomes, and in this regard, an array of viral and host protein interactions must work in tandem. 5′ TAR RNA structures provide extra stability to the unspliced viral mRNA that can interfere with ribosome recruitment and scanning, and for this issue, host helicases are required [414,415,416]. RNA helicase A (RHA) is packaged in virions as well as recruited in the host cell and binds to TAR RNA to enact its helicase functions and promote translation. The depletion of RHA or deficiency of RHA-virion packaging greatly reduces HIV-1 translation levels [416,417,418]. Another helicase, DDX3, has been shown to associate with Tat in cytoplasmic stress granules and is needed for maximal translation [419].

A noteworthy aspect of HIV-1 infection is that the active form of eukaryotic initiation factor 4E (eIF4E) is downregulated by Vpr to induce cell cycle arrest at the G2/M phase [386]. eIF4E is the cap-binding protein component of eIF4F, the protein complex needed for cap-dependent translation [420]. HIV-1 uses the cap-dependent translation route (among others) for its mRNA, so this would seem counterproductive to downregulate eIF4E; however, HIV-1 has workarounds to ensure the translation of its own proteins while halting that of the host. For instance, CBC-bound unspliced viral mRNAs are still actively translated during this cell-cycle arrest, displaying CBC’s ability to circumvent the need for eIF4E [386]. DDX3 also plays a major role when active eIF4E is depleted. DDX3 can act as a substitute for eIF4E by binding the 5′ end of viral mRNA, and through interactions with eIF4G and poly-A-binding protein cytoplasmic 1 (PABP) it can associate with eIF4F to promote translation in cytoplasmic compartments [419,421,422]. Outside of HIV-1, interactions between DDX3 and the CBC to initiate translation have been observed, particularly in regards to cancer progression. However, to our knowledge, there are yet to be any studies directly linking their interactions and translation of HIV-1 [423,424].

Other than cap-dependent translation, HIV-1 also uses internal ribosome entry sites (IRESs) to translate uncapped mRNA [425]. HIV-1 possesses two IRESs, one in the 5′ UTR (HIV-1 IRES), and one in *gag* coding region (HIV-1 Gag IRES) [426,427]. Several IRES Trans-Acting Factors (ITAFs) have been identified as amplifiers of HIV-1 IRES translation. Interestingly, various proteins that assist in the Rev-CRM1-mediated nuclear export of HIV-1 mRNA also play roles as ITAFs: depletion of DDX3, eIF4A, or hRIP would result in decreased translation of uncapped HIV-1 mRNA [428]. hnRNP A1 is a nuclear protein involved in a variety of RNA-processing activities like splicing and export. HIV-1 induces the expression of hnRNP A1 and promotes its accumulation in the cytoplasm where it functions as an ITAF for HIV-1 translation [429]. An HIV-1 IRES pulldown assay identified a variety of proteins that interact with that IRES. PC4, outside of its role in Tat-mediated activation of the 5′LTR, was one of these proteins, revealing a dual role in both transcription and translation [430].

Proteolytic processing of the Gag and Gag-Pol polyproteins is carried out by the HIV-1 protease (PR), which is initially translated as part of the Gag-Pol polyprotein. During or shortly after virion budding, Gag-Pol dimerizes, activating PR’s enzymatic activity, and undergoes “PR precursor autoprocessing”. This releases PR, which subsequently cleaves the packaged Gag and Gag-Pol polyproteins into their individual proteins [421]. Cleavage into the individual proteins, and the sequence in which these cleavage events occur, is required for correct maturation of HIV-1 [422]. The Env (gp160) protein is not part of the Gag or Gag-Pol polyproteins, and is not a substrate of PR, but requires proteolytic processing to be split into the gp120 surface and gp41 transmembrane subunits. Unprocessed or incorrectly processed Env gp160, i.e., improper cleavage into gp120 and gp41, can be directed to lysosomes for degradation or may be assembled into virions and impair the infectivity of the virus [423,424]. For Env proteolytic processing, the host endoprotease furin is utilized by HIV-1 to carry out the cleavage of gp160 into gp120 and gp41, the process of which occurs as gp160 is exiting the Golgi [425,426]. PACS-1 is a cytosolic sorting protein that binds furin and localizes it to the trans-Golgi Network (TGN), thus facilitating furin-mediated cleavage of gp160 [427]. Though important for maximal levels of gp160 cleavage, furin is not essential for the process to occur, as gp160 cleavage was observed in furin-deficient cells [428]. Indeed, furin is not the only protein that can cleave gp160, as another endoprotease, Proprotein convertase 1 (PC1), was also found to mediate gp160 cleavage [429].

Restriction factors against HIV-1 translation employ a variety of approaches to prevent viral protein production. The IFITM family of ISGs, previously mentioned for their anti-entry activities, also impair HIV-1 at the translation step. Here, IFITMs intercept viral mRNA transcripts to prevent their interaction with ribosomes, with transcripts encoding an RRE being the most greatly affected. This disruption is dependent on the intracellular domain of the IFITM, with the N- and C-termini being dispensable [74,431]. HIV-1 requires a specific Gag to Gag-Pol polyprotein ratio, about 20 to 1, for proper replication and infectivity [432]. HIV-1 takes advantage of the Programmed -1 Ribosomal Frameshifting (-1PRF) mechanism of ribosomes, as its mRNA contains the -1PRF signal requirements to induce frameshifting: a “pseudoknot” secondary structure that causes stalling of translation, and a heptanucleotide “slippery” sequence holding the -1 frame for tRNAs bound to the ribosome to fall into. It should be noted that while all mRNAs contain a pseudoknot (and/or other secondary structures) only a small fraction will undergo frameshifting [433,434,435]. During the translation of Gag mRNA, this frameshift will occasionally occur, generating the Gag-Pol polyprotein and ensuring the proper ratio of viral proteins is kept [436,437]. Shiftless (SFL) is a recently described ISG that interferes with the regulation of Gag to Gag-Pol polyproteins in an infected cell. SFL interferes with the translation of both cellular and viral proteins by interacting with mRNA containing the -1PRF signal, as well as actively translating ribosomes, to cause premature termination of translation [437]. Schlafen 11 (SLFN11) is an ISG with a fascinating antiviral effect; SLFN11 can selectively inhibit the translation of HIV-1 proteins, but not global translation, based on the codons used by the virus. HIV-1 displays a bias towards the use of A/T in its genes, namely the rare incorporation of adenine in the third nucleotide position of the codon, making the relatively rare tRNAs that correspond to these codons invaluable to HIV-1 protein synthesis. SLFN11 binds these tRNAs to prevent their use by HIV-1, thus disrupting translation by depleting the available tRNA pool [438,439,440,441]. Schlafen 12 (SLFN12) acts similarly to SLFN11, stalling ribosomes to cause a block in translation in a codon-dependent manner, and has been suggested to play a role in the prevention of reactivation of HIV-1 in CD4+ T cells [442]. HuR, an ELAV-like protein that was previously mentioned as a positive modulator of RT activity, serves to negatively regulate IRES-mediated HIV-1 translation by repressing IRES activity. Despite being an RNA-binding protein, HuR’s influence on IRES-mediated translation does not appear to involve direct binding to the viral RNA [443,444].

Recently, caspase recruitment domain-containing protein 8 (CARD8) was described as an immune sensor for HIV-1 PR activity. As mentioned previously, PR is normally activated and released from the Gag-Pol polyprotein during or after virion budding. Upon infection, this free PR is released into the cytosol and is able to cleave the N-terminal portion of CARD8, releasing the C-terminal subunit that then triggers the caspase-1 inflammasome pathway of pyroptotic cell death. This can occur soon after viral entry, prior to the establishment of productive infection. However, only ~120 copies of PR are packaged within a virion, and PR requires dimerization for its activity. This relatively low abundance of PR early in infection means PR’s interaction with CARD8 is not always guaranteed. In the later stages of infection, premature PR release and activity (i.e., prior to packaging into immature virions) can occur due to overexpression of the Gag-Pol polyprotein or treatment with non-nucleoside reverse transcriptase inhibitors, both of which increase occurrences of Gag-Pol dimerization and thus PR activation, promoting detection by CARD8 in productively infected cells. CARD8 inflammasome activity was originally observed in myeloid cells and was later found in resting, but not activated, CD4+ T cells. Taken together with the observation that bystander resting T cells showing no evidence of productive HIV-1 infection still undergo programmed cell death, CARD8-mediated pyroptosis is suggested to be a contributing factor to the depletion of resting CD4+ T cells prior to the establishment of productive infection [445,446,447,448,449,450,451].

Protein Kinase R (PKR) is an ISG that, upon sensing of viral dsRNA, phosphorylates eIF2α, preventing the formation of the translation initiation complex, thus halting both viral and cellular translation [176]. Interestingly, during HIV-1 infection multiple host factors work against PKR and greatly undercut its antiviral activity. As mentioned previously, PKR can be countered by another ISG, the adenosine deaminase ADAR1, which edits viral RNA and prevents detection by cellular RNA immune sensors [176,452,453]. The TAR RNA binding protein (TRBP), in addition to its transcriptional enhancement role in HIV-1 infection, inhibits PKR in multiple ways. TRBP can bind dsRNA to mask it from PKR, directly bind PKR to block its phosphorylation of eIF4A, and at high concentrations TRBP can also bind with the PKR activator (PACT) to prevent activation of PKR [453,454,455,456,457,458]. Normally, PACT phosphorylates PKR to activate it under conditions of stress; however, during HIV-1 infection PACT is not only sequestered by TRBP to prevent its interaction with PKR but has also been seen to have its role reversed, going from a PKR activator to an inhibitor. Though the exact mechanisms are not fully understood, it may involve PACT binding to ADAR1 [459,460,461,462].

The interruption of translation can also be exploited to the benefit of HIV-1. As mentioned previously, HIV-1 requires a specific Gag to Gag-Pol translation ratio for proper replication and infectivity, and the same is true for the ratio of Gag to Env protein, as Env-deficient viral particles are non-infectious [463,464,465]. During the translation of the Gag polyprotein, the host factor RuvB-like 2 (RVB2) binds the translated MA portion and the 5′UTR of the translating mRNA to enable mRNA degradation. This anti-translation activity is relieved by Env, which competes with RVB2 for MA-binding, reinforcing the hypothesis that RVB2′s mRNA degradation activities serve the best interests of HIV-1 by promoting a proper Gag to Env ratio [466].

Ubiquitination and subsequent degradation via constitutive proteasome (hereafter “proteasome”) or immunoproteasome are the cell’s primary methods of disposing of translated viral proteins and serve a dual role in the immune response. Other than directly halting viral replication in the currently infected cell, proteasome, and immunoproteasome degradation is a key step in presenting HIV-1 antigens by antigen-presenting cells (APCs), like macrophages and dendritic cells, via major histocompatibility complexes I and II (MHC I and MHC II, respectively) [467,468,469,470]. The CA, IN, Tat, Nef, and Vif HIV-1 proteins are all targets of ubiquitination and proteasomal degradation [468,471]. Degradation of Tat by the ubiquitin-proteasome system (UPS) is mediated by the long non-coding RNA NRON. NRON binds Tat and then recruits NRON-binding proteins involved in the UPS, specifically CUL4B, PSMD11, and HUWE1, which facilitates UPS-mediated degradation [472]. HIV-1 CA and IN proteins are both antagonized by the E3-ubiquitin ligase TRIM5α, meting out their proteasome-destined destruction [471,473,474]. Nef is degraded via its interactions with ubiquitin specific protease 15 (USP15), although the exact mechanism is not clear since USP15 usually functions to stabilize proteins by deubiquitinating them [475]. Vif is ubiquitinated by the E3-ligase MDM2, and is noteworthy amongst the HIV-1 accessory proteins as it has a notably short half-life, which may suggest Vif’s intense rates of MDM2-mediated degradation are by design and benefit HIV-1 infection [476,477].

Degradation can also be facilitated outside of the UPS pathway. Lysosomal-associated transmembrane protein 5 (LAPTM5) is a major restriction factor against HIV-1 infection of macrophages and DCs, as it is responsible for transporting Env glycoproteins to lysosomes for destruction [478,479]. Interestingly, another heavy hitter in myeloid defenses is mannose receptor (MR), previously mentioned for its benefits to HIV-1 through its aid in initial virion binding to DCs. Env proteins undergo post-translational processing, such as N-linked glycosylation. One such addition to Env is the mannose patch, and through this region, MR binds Env and Env-containing virions and facilitates their shuttling to lysosomal compartments where they are destroyed [480,481,482].

A consequence of the degradation of HIV-1 proteins is the generation of peptides that can be used by immune cells to generate a large-scale immune response. Antigen-presenting cells such as macrophages and dendritic cells, can present via MHC I to activate CD8+ T cells, or via MHC II to activate naïve CD4+ T cells [27,483,484]. After activation, CD8+ T cells will then kill any infected cell presenting the recognizable antigen via MHC I; nearly all cell types can present via MHC I [485]. Upon activation via MHC II, CD4+ T cells will then differentiate, proliferate, and activate other immune cells to amplify the immune response [486]. Cathepsins are proteases found in endosomes and lysosomes, and play a variety of roles in different immune cells, from processing peptides for MHC II presentation to regulating cytotoxicity of natural killer (NK) and CD8+ T cells [487]. Two cathepsins, D and K, were found to be able to degrade a proteasome-resistant Env gp120 mutant [488]. ER aminopeptidases (ERAPs) are involved in MHC I peptide generation by further degrading proteins post-proteasome. In the case of HIV-1, the p17 MA protein and the p24 CA protein are subject to ERAP degradation [489,490].

## 9. Assembly and Egress

The first step for proper HIV-1 virion assembly and budding, in primary T cells and cell lines, is the localization of virus (and host) components to the peripheral PM, and here the Gag polyprotein’s interactions with host factors are critical [491,492,493]. Localization of Gag to the PM is aided by phosphatidylinositol (4,5) bisphosphate [PI(4,5)P_2_] (PIP2), a phosphoinositide that localizes many host cell factors to the PM [494]. By interacting with basic amino acid residues in the MA domain of Gag, PIP2 binds the Gag polyprotein and transports it to the PM [495]. Assembly sites are not random but are instead sites where lipid rafts, microdomains composed of cholesterol, sphingolipids, and proteins, can be assembled. Cholesterol in particular is a necessity, as depletion of cholesterol permeabilizes the virion membrane and greatly hinders the incorporation of Gag and Pol proteins [496]. Once at the PM, Gag will bind to the PM, an interaction dependent upon its N-terminally myristoylated MA domain [497,498]. Myristoylation of MA of Gag is carried out by N-myristoyltransferase (NMT) and is required for PM association, proteolytic processing, Gag-Gag associations, and budding, but is not necessary for localization of Gag to the PM. PIP2 interaction with MA causes a conformational change exposing its N-terminal myristoyl site, which allows PM binding. This “myristoyl switch” is thought to be a method to prevent aberrant MA binding until PIP2 binds Gag and localizes it to the PM [491,499,500,501,502,503,504,505]. Basic residues in MA, in addition to the N-terminal myristate, also enable binding to the PM-bound acidic phospholipid PS, further enhancing Gag-PM binding stability, while also contributing to the correct formation of MA hexamers needed for proper virion assembly to occur [506,507].

Env proteins must also be localized to the PM for correct assembly to occur. As mentioned earlier, Env gp160 must be properly processed, by furin or PC1, into gp120 and gp41 for localization to the PM to occur, otherwise it will be targeted to lysosomes where it will be degraded [508]. Post-cleavage, gp120 and gp41 form a complex, which in turn trimerizes with other gp120/gp41 complexes to form an Env complex. gp41 has a long cytoplasmic tail (CT) of around 150 amino acids, and this CT is what interacts with host factors to facilitate Env complex localization to the PM. Rab11-FIP1C/RCP (FIP1C) is from a family of cargo-sorting proteins and, together with its binding partner Rab14, interacts with Env complexes in a CT-dependent manner to localize them to the PM and incorporate Env into the virions [509,510]. Adaptor Protein 1 (AP-1) also appears to play a role in the sub-cellular localization of Env after it exits the TGN, and as with FIP1C and Rab14, the gp41 CT is the facilitator of interaction [511]. Interactions between the gp41 CT and MA also contribute to the proper incorporation of Env into budding virions, as mutations in MA or the CT lead to inefficient incorporation of Env into the viral particles [498,512,513].

In myeloid cells, assembly was observed not only at the peripheral PM but in cytosolic compartments [514]. Originally these were thought to be late endosomes or multi-vesicle bodies, but more recent studies have since shown these compartments to be internalized sections of the PM, marked by the tetraspanins CD9, CD53, and CD81, and are still connected to the cell surface by “virion channels”, tubules only slightly bigger than the virions themselves; these sites were dubbed virus-containing compartments (VCCs) [515,516,517,518,519,520,521]. VCCs are seen during HIV-1 infection, but the infection itself is not necessary for VCC formation. The surface lectin Siglec-1, found on macrophages and DCs, was seen to be crucial for the formation of VCCs following binding to the gangliosides on virus-like particles, independent of subsequent productive infection [522]. These VCCs also mediate the cell-to-cell transfer of HIV-1 through virological synapses, mentioned earlier in this review [65,523]. Interestingly, the restriction factor tetherin (discussed more later) plays a role beneficial to HIV-1 when it comes to VCC formation, as tetherin-KO macrophages showed impaired formation and distribution of VCCs [524]. As in T cells, PIP2 plays a role in localizing Gag to VCCs, which is consistent with VCCs being comprised of internalized PM. Similarly, FIP1C and Rab14 are needed for localization of Env complexes to VCCs [509,525,526].

Once at the PM, Gag sequesters lipid rafts and tetraspanin-containing microdomains to serve as platforms for HIV-1 particle assembly [527,528]. Capsid assembly is aided by ATP-binding cassette E1 (ABCE1, or HP68, or RLI), a transporter protein that binds Gag through the NC domain, though it does not stay attached post-assembly [529,530,531]. Gag-Gag multimerization at the PM is regulated by cellular STAU1 through interactions with the two zinc fingers (ZFs) of Gag’s NC domain. STAU1 also binds genomic RNA, and through this association is packaged into virions [412,532,533,534]. Inositol hexakisphosphate (IP6) also promotes interactions between Gag molecules and has a conserved role in many lentiviruses. IP6 interacts with the CA domain of Gag to promote assembly of the immature Gag lattice, which then undergoes proteolytic cleavage, after which IP6-CA interactions facilitate formation of the mature capsid structure [535,536]. Gag and Gag-Pol polyproteins are packaged in a radial manner, with the MA domain facing outwards and their C-terminus towards the center [492]. Full-length viral RNA dimers that will serve as the HIV-1 genome are recognized in the cytoplasm by the NC domain of Gag, prior to PM localization, and packaged into the assembling virion [537,538,539]. Nucleolin, an RNA binding chaperone protein that plays a role in ribosome assembly, binds the genomic RNA of HIV-1 and forms a complex with Gag through NC interactions. This leads to virion packaging of both nucleolin and RNA, the latter of which contains the *cis*-acting RNA element psi, a packaging signal, and enhances virion budding through Gag recognition of psi [540,541,542]. Accessory proteins Vpr, Vif, and Nef are packaged in the virion, with the latter two later undergoing intravirion processing by PR [543,544,545,546,547]. Through a variety of methods, such as mass spectrometry, immunomagnetic capturing, and proteomic profiling, dozens of host proteins have been identified as being incorporated into the virion core or envelope (reviewed in [548,549]). Most packaged proteins are those that beneficially impact a step, or multiple, of the viral replication cycle. Examples already mentioned in this review include PSGL-1, CypA, IP6, UPF1, HDAC1, Staufen, RHA, ADAR1, and CBP80 [84,105,118,163,167,386,418,534,550]. Unfortunately for HIV-1, restriction factors such as APOBEC and MOV10 are also incorporated into virions, where they can continue to wreak havoc after the virion has departed from the host cell [208,226].

With the HIV-1 genome, HIV-1 proteins, and host proteins in the process of being packaged, the final act of budding proceeds. Once again, Gag interactions with host proteins ensure this process is successful. p6 of the Gag polyprotein contains amino acid motifs dubbed “late domains” (L-domains) that were found to be critical to completing separation of the virion from the cell. Deletions of p6 or mutations in this motif would lead to incomplete budding, with virions attached to host cells by thin “stalks” [551,552]. It was discovered that p6 L-domains interact with proteins of the endosomal sorting complex required for transport (ESCRT) pathway at the site of virion assembly to facilitate proper budding [553,554]. The ESCRT pathway is composed of various ESCRT protein complexes (ESCRT-0 to ESCRT-III), and is a trafficking pathway used by cells to sort ubiquitinated membrane proteins via multivesicular bodies (MVBs) to lysosomes for destruction [554,555]. An L-domain of p6 with the PTAP amino acid motif was found to be able to bind the ubiquitin enzyme 2 variant (UEV) region of tumor susceptibility gene 101 (TSG101), a protein of the ESCRT-I protein complex, and recruit TSG101/ESCRT-I to budding sites to initiate ESCRT assembly. Consistent with previous reports that UEV’s bind Ub, ubiquitination of p6 increases its binding affinity with TSG101′s UEV. To the same point, a Gag construct with a deubiquitinating enzyme fused on had impaired TSG101 binding, suggesting ubiquitination of Gag is a key step in efficient recruitment of TSG101. As noted above, anomalous or nonexistent p6 L-domains heavily impairs budding. Likewise, depletion of TSG101 attenuates budding but can be rescued once TSG101 is re-introduced [556,557,558,559,560,561,562]. Once bound to Gag and localized at the PM, TSG101/ESCRT-I can recruit ESCRT-II through interactions with ESCRT-II subunits EAP30 and EAP45. EAP20 of ESCRT-II can then recruit ESCRT-III through interactions with charged multivesicular body protein 6 (CHMP6), a subunit of ESCRT-III that can bind to vacuolar protein sorting-associated protein 28 (VPS28) of ESCRT-I. Interestingly, siRNA-mediated knockdown of ESCRT-II component EAP20, or CHMP6 of ESCRT-III, did not significantly impact HIV-1 release or infectivity, implying that HIV-1 budding is not ESCRT-II-dependent [563,564,565,566].

A second L-domain of p6, YPXL (where X is any amino acid), binds the V region of apoptosis-linked gene 2 (ALG-2)-interacting protein (ALIX). ALIX can interact with TSG101/ESCRT-I, as well as with CHMP4 proteins of the ESCRT-III complex to facilitate ESCRT-III recruitment independently of TSG101. Interactions with CHMP4 are mediated through ALIX’s Bro1 domain and are regulated by the ATPase VPS4. Impairment of ALIX/p6 interactions is less catastrophic for HIV-1 budding than impaired TSG101/p6 interactions, indicating that the ALIX route of ESCRT-III recruitment is an approach useful in specific scenarios, e.g., cells under-expressing TSG101 or an HIV-1 mutant without a PTAP motif. Interestingly, ALIX-depleted cells showed multiple recruitments of full ESCRT complexes to budding sites, a rarity in wild type scenarios, suggesting that p6/ALIX interactions may facilitate or coordinate correct recruitment of ESCRT machinery [564,567,568,569,570]. The Bro1 domain of ALIX can also interact with the NC domain of Gag, via basic residues in the N-terminal region and ZFs of NC, linking Gag to ESCRT-III in a p6-independent manner [571,572].

Another host factor Gag interacts within the course of virion budding is angiomotin (AMOT), an angiostatin-binding protein. HIV-1 variants without PTAP and YPXL L-domains, or lacking a p6 domain altogether, can still bud efficiently when the HECT ubiquitin E3 ligase NEDD4L is overexpressed. While other retroviruses can bind NEDD4L directly through a PPXY L-domain, HIV-1 Gag lacks this domain. AMOT serves as the connection between Gag and NEDD4L, as it can bind somewhere in residues 278-377 of Gag and bind NEDD4L through AMOTs PPXY motifs and NEDD4L’s WW domains. Depletion of AMOT or NEDD4L impairs the budding of HIV-1 strains with Gags lacking p6 at an early stage in the process. Despite these strains being unable to directly bind TSG101, NEDD4L-mediated rescue of the strains requires the presence of TSG101, suggesting that NEDD4L acts upstream of TSG101 and connects PTAP-and-YPXL-defective mutants to the ESCRT pathway [573,574,575].

Whether through TSG101, ALIX, or AMOT, the end of Gag’s interactions with these host proteins results in the recruitment of ESCRT-III components to the site of the budding virion (interactions summarized in Figure 2). ESCRT-III is critical for membrane fission during the cytokinesis phase of mitosis and is similarly essential for complete separation of budding virions from the host cell [576]. Of the 12 ESCRT-III-like proteins expressed by humans, only CHMP2 and CHMP4 appear to be indispensable for HIV-1 budding. As noted previously, ALIX recruits ESCRT-III through CHMP4; CHMP2 and CHMP4 also interact directly with each other, even in the absence of other ESCRT-III components, and both can bind and recruit VPS4 [567,577]. VPS4 is an AAA-type ATPase whose enzymatic activity is required for disassembly and recycling of membrane-bound ESCRT-III complexes and is the mediator between the ALIX/CHMP4 interactions. Interruption of the CHMP2-CHMP4 or CHMP2/4-VPS4 interactions impairs HIV-1 budding [567,576,578,579,580]. Like so many other host proteins, ESCRT-I, ESCRT-III, ALIX, and VPS4 are all incorporated into the virion during the budding process [492]. As mentioned above, host proteins of the PM can be incorporated into the virion as it is budding, with some, such as ICAM-1, playing a role for that virion/virus during the next round of infection and replication. Other host surface proteins taken by the virion during budding include MHCI and MHCII, CD40, LFA-1, and many more [66,68,549].

The host cell has many countermeasures to prevent the proper assembly and budding of HIV-1 progeny virions. The Ras GTPase-activating-like protein IQGAP1 (IQGAP1, or p195) is a scaffolding protein that serves as a regulator of actin-cytoskeleton rearrangements, as well as many signaling pathways [581,582]. IQGAP1 pre-emptively restricts virion assembly by binding to NC or p6 of Gag, independent of the L-domains, and preventing Gag from assembling at the PM [583]. The ISG TRIM22 functions similarly, binding Gag and disrupting its trafficking to the PM [584]. Another ISG, ISG15, impairs assembly at multiple steps of the ESCRT recruitment process: ISG15 inhibits both Gag and TSG101 ubiquitination and prohibits interaction between Gag’s PTAP and TSG101, preventing TSG101-mediated recruitment of the rest of ESCRT-I; ISG15 also binds CHMP5 of ESCRT-III, and through this disrupts VPS4 recruitment by blocking VPS4 interactions with its coactivator protein LIP5; and ISG15 can be conjugated onto Gag by the host factor HERC5 to disrupt an early stage in Gag assembly [585,586,587]. PSGL-1, an ISG previously mentioned for its restrictions at the fusion and reverse transcription steps in HIV-1 replication, also impairs HIV-1 budding. PSGL-1 can bind gp41 to restrict Env incorporation into budding virions, reducing subsequent infectivity of the virion as a result. PSGL-1 also interacts with Gag to facilitate its incorporation into the virion, leading to its anti-fusion activities mentioned above [84,221,222]. The T-cell immunoglobulin (Ig) and mucin domain (TIM) family of transmembrane proteins play roles in intracellular signaling, with different TIM proteins being more important in T cells and/or macrophages, e.g., TIM-1 in activated T cells, TIM-4 in myeloid cells, TIM-3 in both [588,589,590]. TIM proteins can all bind the PM-bound PS, which as mentioned earlier is important for Gag-PM binding and MA hexamer assembly. The binding of PS by TIM-1, TIM-3, or TIM-4 results in the inhibition of virion release, retaining the virions at the PM of T cells, with similar effects observed with TIM-3 in MDMs [591]. TIM-1 also appears to be stabilized by SERINC5, a restriction factor at the fusion step of the HIV-1 replication cycle (see Section 2 [592].

In response to HIV-1 infection or stimulation by interferon, viperin is an ISG highly induced in macrophages, relative to the low levels of induction observed in monocyte-derived DCs and CD4+ T cells. Viperin combats HIV-1 infection at the assembly stage by disrupting stable formation of the lipid rafts Gag coalesces to serve as virion assembly sites. However, these antiviral effects are only substantial against certain HIV-1 strains [593,594,595]. GBP5 and GBP2 are ISGs, specifically GTPases, involved in intrinsic immune cell activation, are highly stimulated in MDMs by type I and II IFNs, and can be stimulated in T cells by type II IFN. GBP5 and GBP2 interfere with the HIV-1 Env glycoprotein by localizing at the Golgi and deactivating furin, the host protease that cleaves gp160 into gp120 and gp41. This impairs gp120 trafficking to the PM, increasing virion incorporation of unprocessed gp160 and decreasing that of gp120 This in turn leads to reduced infectivity of progeny virions due to their lack of gp120. Interestingly, HIV-1 strains with defective *vpu* genes have higher resistance to GBP5, though this is not yet confirmed for GBP2. This is likely due to Vpu’s ability to downregulate NF-кB to restrict ISG expression, as NF-кB also acts as a transcription factor for HIV-1, and thus Vpu-defective mutants will have higher expression of Env that can overwhelm GBP5′s antiviral effects. This may be an explanation for the relatively high frequency of M-tropic strains with a defective *vpu* gene [596,597,598,599,600]. MARCH8 is a RING-finger E3 ubiquitin ligase highly expressed in MDMs and DCs that normally functions to downregulate host transmembrane proteins, such as MHCII, and directly impacts viral infectivity by sequestering Env proteins from the cell surface, leading to virions lacking Env glycoproteins and thus being noninfectious. Interestingly, despite its normal functions involving UPS-degradation of cellular proteins, MARCH8′s antiviral activity does not alter Env protein levels, revealing that MARCH8-mediated downregulation of Env does not involve proteasomal degradation [601,602,603,604]. From the same protein family of E3 ligases, MARCH1 and MARCH2 have similar anti-Env activities to MARCH8; unlike MARCH8, MARCH1 and MARCH2 are stimulated by type I IFNs, and MARCH2 is upregulated by HIV-1 infection alone [604,605,606].

One of the most extensively studied restriction factors is bone marrow stromal antigen 2 (BST-2), also known as tetherin. Tetherin is constitutively expressed in many cell types but is strongly induced by type I IFN, thus it is considered to be an ISG. Tetherin is a transmembrane protein that is found in foci dotted across the PM, and as the name implies, tetherin’s antiviral activity is the tethering of virions to the outside of the plasma membrane, as well as to each other, after they bud. This is accomplished by the incorporation of several dozen tetherin homodimers into the virion, with a preference for C-terminal insertion into the virion in an axial manner, as the C-terminus contains a glycosyl-phosphatidylinositol (GPI) anchor. The cytosolic N-terminus of tetherin, and coiled-coil structure adopted by the tetherin homodimer, are critical for retaining virions on the cell surface. As the virion with tetherin’s inserted C-terminus attempts to leave, the N-terminus will stay rooted in the PM of the host cell, thus tethering the virion to the cell [607,608,609,610,611,612,613]. However, this retention of virions may actually benefit HIV-1, as it can facilitate the formation of virological synapses and cell-cell transfer of the virions [614]. Finally, tetherin also functions as a sensor for viral infection: through its cytosolic domain, tetherin acts as an immune sensor that stimulates NF-кB in response to virion binding, leading to type I IFN production [615,616].

## 10. Concluding Remarks

Since the identification of HIV as the pathogen responsible for Acquired Immunodeficiency Syndrome (AIDS) in the early 1980s, the list of host factors identified as playing a role in its replication cycle has only continued to grow [617]. New factors and new interactions between these factors and HIV, or between the factors themselves during HIV infection, are constantly being discovered and explored further. A wide assortment of techniques potentiates these discoveries: co-immunoprecipitation, yeast-two-hybrid assays, fluorescent microscopy and imaging, mass spectrometry, RNA-hybridization, genome-wide association studies, CRISPR, RNA interference; these methods, and more, have been successful in elucidating the mysteries of what host factors HIV interacts with, and how [618,619,620,621,622,623,624,625,626]. As the complex web of dynamics between all factors that help or hinder HIV over the course of infection grows, so too does our sum-total knowledge of this viral scourge. Every factor identified, interaction studied, and mechanism understood brings us one step closer to our goal: a true and final cure for HIV and AIDS.

## Figures and Tables

**Figure 1 viruses-16-01281-f001:**
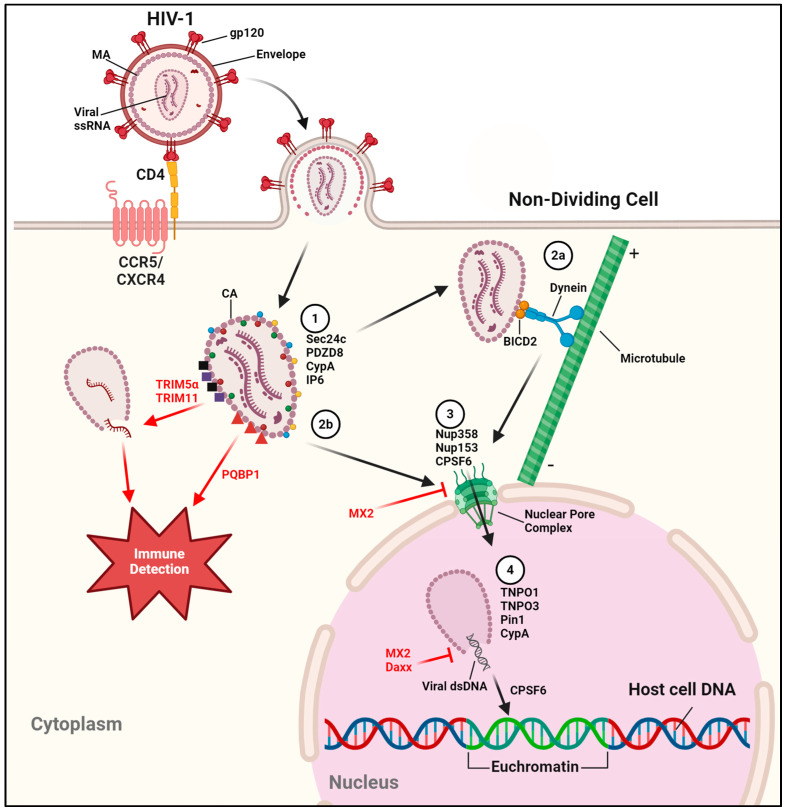
Host factors and restriction factors interact with the HIV-1 capsid to either assist or prevent successful cytoplasmic transport, nuclear entry, and capsid uncoating. (1) After attachment and envelope fusion with the plasma membrane, the capsid is released into the cytoplasm. Factors found within the capsid (CypA, red spheres; IP6, green spheres) as well as outside of the capsid (Sec24c, yellow spheres; PDZD8, blue spheres) bind the capsid to promote stability during cytoplasmic transport. Restriction factors (TRIM5α, black squares; TRIM11, purple squares; PQBP1, red triangles) facilitate capsid disassembly and/or immune detection of the viral genome. Transport of the capsid can occur in a microtubule-dependent (2a) or microtubule-independent (2b) manner. Using the adaptor protein BICD2, the microtubule-associated protein dynein moves the capsid inwards towards the nucleus; FEZ1 and kinesin-1 (not shown in Figure), though outwards-bound from the nucleus, still positively impact capsid localization to the nucleus. (3) At the nucleus, capsid entry through the nuclear pore complex is aided by host factors (Nup358, Nup 153, CPSF6) and prevented by restriction factors (MX2). (4) Once inside the nucleus, host factors promote capsid destabilization and disassembly (TNPO1, TNPO3, Pin1, CypA), ensuring proper viral genome release, followed by localization to sites of open chromatin by CPSF6. At the same time, restriction factors (MX2, Daxx) will attempt to prevent uncoating and genome release.

**Figure 2 viruses-16-01281-f002:**
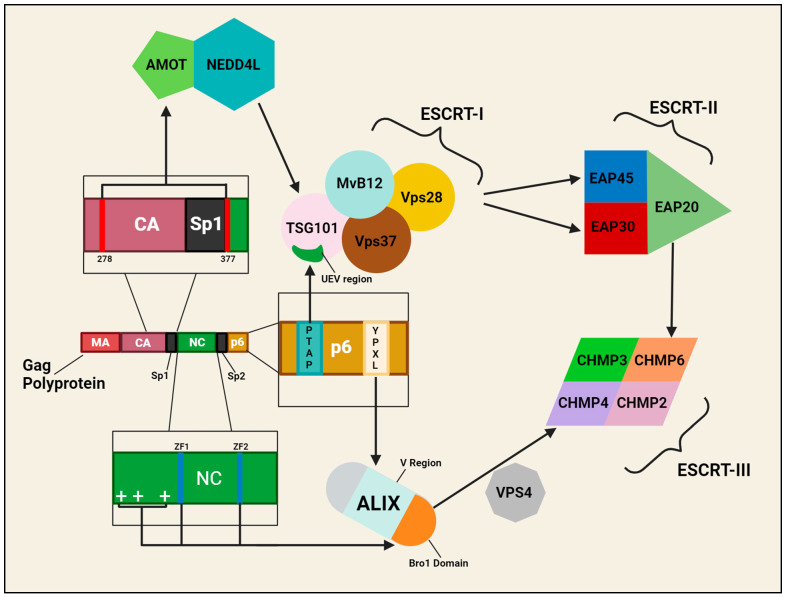
Various components of the Gag polyprotein interact with host factors to facilitate the recruitment of ESCRT pathway machinery, leading to productive and efficient virion budding. See text for further details.

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
