# Peer review of "Help or Hinder: Protein Host Factors That Impact HIV-1 Replication"

_viruses, 2024, doi:10.3390/v16081281_

Round 1

Reviewer 1 Report

Comments and Suggestions for Authors

                This manuscript is a literature review on host proteins that have been shown to be involved with normal infection processes during the HIV-1 virus replication cycle. To say that this review is comprehensive would be an understatement, as suggested by the 611 references that are used. The review is well organized, starting with virus entry and moving through the normal life cycle, ending with budding and maturation. It is also a well written analysis and something that will be used as a general reference for years to come. Bravo to the authors! The only minor criticism is that it would have been nice if these experts included some editorial opinion (or even a separate short section) on what they thought were best host targets for antiviral development for therapy and/or cure.

Minor Comments

1.       Line 66: R5X4 viruses are also called dual-tropic viruses and this should be included.

2.       Many interferon stimulated genes produce proteins that are used to protect uninfected cells rather than the infected cell that makes the protein. The authors do not make this distinction in the review. Perhaps a sentence can be added to address this.

3.       Line 609: Grammar; it’s written as if Rev is an mRNA and not a protein.

4.       Starts at line 702: The discussion on the -1PRF mechanism makes it seems like only a small proportion of mRNAs have this “pseudoknot”. In fact, all the mRNAs have this, but only a small proportion induce the frameshift. This should be clarified.

5.       Paragraph on CARD8 (Line 723) needs further clarity. If premature protease activation occurs prior to assembly/budding (and must wait until polyprotein is produced after integration), how can it then function (line 729) “soon after viral entry, as the aberrant PR is released into the cell, but prior to establishment of productive infection?”

Author Response

Thank you for reviewing our paper and for your comments and suggestions. Please see the below responses.

Comment: The only minor criticism is that it would have been nice if these experts included some editorial opinion (or even a separate short section) on what they thought were best host targets for antiviral development for therapy and/or cure.

Response: Thank you for this criticism, we do agree a discussion on host targets for antiviral development for therapy/cure is warranted, and was originally planned to be included but ultimately cut (along with other planned sections). Unfortunately, in order to comply with time and author availability constraints, we decided to keep our focus on providing established information rather than sharing our own hypotheses. These constraints are still in effect, and so we will be unable to provide an opinion on host targets with the extensive thought and consideration it deserves.

 Responses to numbered Comments:

Comment 1: Line 66: R5X4 viruses are also called dual-tropic viruses and this should be included.

Response 1: We agree that the term "dual-tropic" should have been mentioned originally, and has now been included (Line 66).

Comment 2: Many interferon stimulated genes produce proteins that are used to protect uninfected cells rather than the infected cell that makes the protein. The authors do not make this distinction in the review. Perhaps a sentence can be added to address this. 

Response 2: Thank you for this comment, it has prompted us to add a statement in the "Reverse Transcription" section about how interferons released by infected cells can prime uninfected cells for the incoming virus. We also added a statement addressing direct release of ISGs that protect uninfected cells. Please consider the sentences below:

Starting at Line 322: “Regardless of the RNA/DNA sensing pathway, the end result is expression and release of interferons that in turn promote expression of antiviral ISGs that counter viral infection in infected cells. Secreted interferons can also bind to interferon receptors of neighboring uninfected cells, priming them for the oncoming virus, inducing expression of ISGs prior to infection. In addition, some ISGs, such as IFITM and ISG15, can be released from the infected cell to uninfected cells, directly acting as defenses to the incoming virus [197–199].”

Comment 3: Line 609: Grammar; it’s written as if Rev is an mRNA and not a protein.

Response 3: Agreed, wording was unclear. We have modified the sentence's wording to make clear that Rev is a protein. (Lines 613-614, our version of the manuscript appears to have slightly different Line numbering than yours)

Comment 4: Starts at line 702: The discussion on the -1PRF mechanism makes it seems like only a small proportion of mRNAs have this “pseudoknot”. In fact, all the mRNAs have this, but only a small proportion induce the frameshift. This should be clarified.

Response 4: Thank you for bringing this fact to our attention. We have added further detail that the -1PRF mechanism requires a -1PRF signal comprised of both the pseudoknot and a slippery sequence. We noted that all mRNA have the pseudoknot and/or other secondary structures but only a fraction will undergo frameshifting.

Starting at Line 702: “HIV-1 requires a specific Gag to Gag-Pol polyprotein ratio, about 20 to 1, for proper replication and infectivity [429]. HIV-1 takes advantage of the Programmed -1 Ribosomal Frameshifting (-1PRF) mechanism of ribosomes, as its mRNA contains the -1PRF signal requirements to induce frameshifting: a “pseudoknot” secondary structure that causes stalling of translation, and a heptanucleotide “slippery” sequence holding the -1 frame for tRNAs bound to the ribosome to fall into. It should be noted that while all mRNAs contain a pseudoknot (and/or other secondary structures) only a small fraction will undergo frameshifting [430–432].”

Comment 5: Paragraph on CARD8 (Line 723) needs further clarity. If premature protease activation occurs prior to assembly/budding (and must wait until polyprotein is produced after integration), how can it then function (line 729) “soon after viral entry, as the aberrant PR is released into the cell, but prior to establishment of productive infection?”

Response 5: Thank you for pointing out the lack of clarification. Further study into this topic has provided us with a better understanding of PR activation of CARD8, and led us to revise and reword the paragraph. 

Starting at Line 723: "Recently, caspase recruitment domain-containing protein 8 (CARD8) was described as an immune sensor for HIV-1 PR activity. As mentioned previously, PR is normally activated and released from the Gag-Pol polyprotein during or after virion budding. Upon infection, this free PR is released into the cytosol and is able cleave the N-terminal portion of CARD8, releasing the C-terminal subunit that then triggers the caspase-1 inflammasome pathway of pyroptotic cell death. This can occur soon after viral entry, prior to establishment of productive infection. However, only ~120 copies of PR are packaged within a virion, and PR requires dimerization for its activity. This relatively low abundance of PR early in infection means PR’s interaction with CARD8 is not always guaranteed. In the later stages of infection, premature PR release and activity (i.e. prior to packaging into immature virions) can occur due to overexpression of the Gag-Pol polyprotein or treatment with non-nucleoside reverse transcriptase inhibitors, both of which increase occurrences of Gag-Pol dimerization and thus PR activation, promoting detection by CARD8 in productively infected cells. CARD8 inflammasome activity was originally..."

Thank you again for reviewing our paper, we hoped to have sufficiently addressed your comments. A fully revised version with all reviewers’ comments addressed will be submitted separately.

Reviewer 2 Report

Comments and Suggestions for Authors

In this review, authors provide an excellent overview of the host factors involved in HIV-1 replication. They highlight the restriction factors or host dependency factors involved in every step of the virus replication.  It will be a go to review to get an understanding of the factors involved in HIV-1 replication. Amount of information provided is pretty impressive.

Minor comments: 

Line 30: spelling error “enveloped”

Line 47. Missing period after the references

Line 54-55: Clarify productive infection of dendritic cells

Line 108-110: sentence is not clear

Line 698 to 700: Relation to the “role in the host immune response” in this sentence is not clear

Line 818: Two “not” in the same sentence. Check the error. 

Long sentence structures with semicolons makes it difficult to read and understand. Sentence construction needs to be improved.  

Comments on the Quality of English Language

Long sentence structures with semicolons makes it difficult to read and understand. Sentence construction needs to be improved.  

Author Response

Thank you for taking the time to review our paper. Please see the below responses to your comments, with changes or additions denoted in red font. 

Comment 1: Line 30: spelling error “enveloped”

Response 1: Thank you for catching this. Corrected to "envelope".

Comment 2: Line 47. Missing period after the references

Response 2: Thank you for pointing this out. Period added.

Comment 3: Line 54-55: Clarify productive infection of dendritic cells

Response 3: Thank you for this comment. We consider productive infection to be infection that results in creation of infectious progeny, and agree that for clarification the definition should be added. This definition has been added to Lines 46-47 where "productive infection" is first mentioned. Since dendritic cells have been seen to be infected by HIV-1 and to produce infectious progeny virions after infection, albeit to a lower extent than other HIV-1 target cells, we believe they fit the definition. Their lower levels of productive infection are mentioned in Line 93. 

Comment 4: Line 108-110: sentence is not clear

Response 4: Thank you for this comment. We agree the wording was wordy, confusing, and implied a role for AP-2 in endocytosis of HIV-1 rather than envelope fusion. Consider the revised sentence below

Line 108-110: “Adaptor Protein Complex 2 (AP-2) is a heterotetrameric protein found on the PM involved in endocytosis of the host cell’s surface receptors, and plays a role in HIV-1 envelope fusion.”

Comment 5: Line 698 to 700: Relation to the “role in the host immune response” in this sentence is not clear

Response 5: Thank you for bringing this to our attention. Upon reviewing the reference for the statement, we see our interpretation was in error. The proposed relation has been removed and replaced with correct information from the same references.

Line 698-700: “Here, IFITMs intercept viral mRNA transcripts to prevent their interaction with ribosomes, with transcripts encoding an RRE being the most greatly affected. This disruption is dependent on the intracellular domain of the IFITM, with the N- and C-termini being dispensable.”

Comment 6: Line 818: Two “not” in the same sentence. Check the error. 

Response 6: The second "not" was in error and has been removed.

Comment 7: Long sentence structures with semicolons make it difficult to read and understand. Sentence construction needs to be improved.  

Response 7: Thank you for this suggestion. Overuse of semicolons has been corrected, and the affected sentences were reworded or rearranged for clarification.

Thank you again for reviewing our paper. A fully revised version with all reviewers’ comments addressed will be submitted separately.

Reviewer 3 Report

Comments and Suggestions for Authors

The manuscript by Moezpoor and Stevenson seeks to describe cellular factors Involved in HIV replication. This is an enormous task, given that the number of such factors is huge and is increasing every day. The authors did a good job describing many of them, involved at different steps of HIV life cycle. Due to the number of the involved factors, it is practically impossible to to provide a deep mechanistic analysis, so in most cases the authors restrain to the description of the function, rather than deep mechanistic analysis. As such, the manuscript has value as an encyclopedia of currently known factors, which should be regularly updated to stay current.

There are several things that need to be corrected to improve the flow of the narration.

1. The authors should correct grammatical errors, such as missing periods, plural vs singular, etc.

2. They should be more precise when describing specific features of the virus. Specifically, when talking about the HIV genome, they mention LTRs. LTRs appear only in the proviral DNA, whereas genomic RNA has short 5' and 3' repeats at the ends, which are make up the LTRs upon reverse transcription.

3. They mention the role of cholesterol, but fail to describe specific factors that regulate the cholesterol homeostasis and thus affect HIV replication. In particularly, the abundance of ABCA1 has been shown to affect HIV replication (PMID #26126533).

Comments on the Quality of English Language

Minor grammatical edits are required.

Author Response

Thank you for reviewing our paper and for your comments and suggestions. Please see the below responses.

Comment 1: The authors should correct grammatical errors, such as missing periods, plural vs singular, etc.

Response 1: Thank you for this comment; we have attempted to go through and correct grammatical errors where spotted.

Comment 2: They should be more precise when describing specific features of the virus. Specifically, when talking about the HIV genome, they mention LTRs. LTRs appear only in the proviral DNA, whereas genomic RNA has short 5' and 3' repeats at the ends, which are make up the LTRs upon reverse transcription.

Response 2: Thank you for bringing this oversight to our attention. We have corrected this error, using UTRs when on the topic of the HIV RNA genome, and LTRs once the genome has been reverse transcribed.

Comment 3: 3. They mention the role of cholesterol, but fail to describe specific factors that regulate the cholesterol homeostasis and thus affect HIV replication. In particularly, the abundance of ABCA1 has been shown to affect HIV replication (PMID #26126533).

Response 3: Thank you for the suggested addition of ABCA1 to our list of factors. We have added a brief discussion of ABCA1 to our "Fusion" section, as its impact on virion infectivity is perfectly suited for this section, particularly the final paragraph where we discuss factors that affect fusion after HIV-1 has already infected and budded from a cell. Unfortunately, due to recent time and resource constraints on the authors, we will be unable to delve into further research on these cholesterol-controlling factors and include more in our review, but feel that ABCA1 is an excellent representative of these factors and thank you for reference provided. (Starting from Line 157):

"As implied by the antiviral activity of IFITMs, cholesterol homeostasis is a key element of successful fusion. The cellular cholesterol transporter ATP binding cassette transporter A1 (ABCA1), also known as cholesterol efflux regulatory protein (CERP), controls efflux of cholesterol from cells. This is to the detriment of HIV-1, as lower cellular cholesterol levels reduces cholesterol incorporation into budding virions, which has been demonstrated to negatively impact infection at the stage of fusion. Nef-deficient HIV-1 strains are particularly vulnerable to ABCA1’s effects, as Nef suppresses ABCA1 function in CD4+ T cells and macrophages by inducing its lysosomal degradation [85–88]."

Thank you again for reviewing our paper, we hoped to have sufficiently addressed your comments. A fully revised version with all reviewers’ comments addressed will be submitted separately.

Reviewer 4 Report

Comments and Suggestions for Authors

The manuscript titled: Help or Hinder: Protein Host Factors That Impact HIV-1 Replication by Moezpoor and Stevenson provides a very comprehensive discussion of all known interactions between HIV-1 and host cell proteins. It should provide a very useful starting point for those learning about HIV-1 for the first time and for those seeking to catch up on the present knowledgebase. While this is a nice review, there are several points throughout that should be fixed to eliminate confusion for those new to the field and to maintain the best level of accuracy.

A general point is that there’s a substantial degree of anthropomorphizing of the virus. I will highlight this but may not have caught all instances. Also, in various places HIV-1 is referred to as HIV-1-1. This may have been a result of searching and replacing but should of course be fixed.

The abstract reads like a justification rather than an abstract.

L22: HIV’s consider changing HIV possessives throughout.

L30: should be envelope rather than enveloped

L31: consider “producer cell”

L34: technically, the genome, the RNA in the virion, is not flanked by Long Terminal Repeats as the 5’ end has R and U5 and the 3’ end has U3 and R.

L34: consider revising the wording referring to the lipid bi-layer envelope versus the envelope glycoprotein. This may be difficult but will avoid confusion for those new to HIV-1.

L43: consider adding “on” for “than on myeloid” for better parallel structure.

L61-62: the conformational change allows the binding. As stated here it looks like the conformational change itself is doing the binding.

L89: should be “ the low level” (singular)

L90: should be CD4 binding, no hyphen.

L111: glycine should be lower case

L126: is fusion really the first chance to avoid infection? What about binding? This should be restated. I’m sure it’s not meant to be a source of discussion.

Paragraph starting L126. Maybe the proposed IFITM mechanism(s) should be discussed first

L150: envs of some strains?

L159: released into the cytosol?

L165: consider changing “may be  transported” to “is transported” to reduce the multiple words reflecting doubt.

L169: “into the nucleus” rather than “to the nucleus”

L181: should be “that regulates”

Figure 1: The title should be revised because negative factors are being highlighted as well.

L250: “RT to create viral cDNA” should be revised to reflect that the cDNA is transcribed from the RNA template. Create suggests a less structured process.

L252: RNAse H abilities should be revised

L293: create should be considered for revision, maybe provides more targets.

L320: consider revising “work to overcome”

L323: consider whether it has been resolved whether APOBEC 3 binds viral RNA or could bind other RNA.

In the APOBEC3G discussion it might be informative to include the G to A transition in the viral coding sequences.

L362-3: examples of HIV1-1.

L386: please consider revising HIV-1’s next goal”

L395: please consider revising “quality location” to be more informative.

L398: consider whether there are really rates

L400: consider whether the comma between transcription and euchromatin is correct.

L402: consider whether changing to provirus formation.

L411 and L414: need the quotation marks if indeed tethering is going on etc.

L460: is the hyphen between the serine and the 2 is required. One paper used this designation, but most don’t.

L470: perhaps “whether it’s necessary has not been resolved”

L474: consider another option for “unbinds:

L476: Please consider another option for “interweaving”

L490: there are better options for “tricked” 

L527: what’s meant by “overlapping host factors” is not clear. Perhaps HIV-1 transcription in … uses many of the same factors…”

L536: “double edged sword” and “for better or worse” cliché’s might be avoided.

L574: Lysine shouldn’t be capitalized

L580: “Unfortunately” is unnecessary here

L609: the fully spliced RNA isn’t Rev, but rather encodes Rev.

L609 and 613 has Rev RETURN to the nucleus when Rev was just translated in the cytoplasm. Here Rev enters the nucleus (for the first time).

L610: Here “rescue” is too imprecise. If the initial highly spliced messages were “rescued” the virus would not function, so this is a relative term. How about “Rev allows a shift from highly spliced to less spliced messages”

L813: Critical necessity is redundant.

L814: “impacts incorporation” is too vague “hinders incorporation” perhaps?

L826: to allow virus assembly to occur?

L838 “MA of Gag” consider revising to just MA

L931: please consider a more specific term than defunct.

L964: if we’re talking about a cycle than there is really no final stage. Please consider revising this terminology.

L991: It’s not clear what you mean by “greatly upregulated.” Is it highly expressed or upregulated in response ti interferon?

L1030 and 1032: twist of irony and watchdog may be out of place here.

Comments on the Quality of English Language

Too much anthropomorphizing and a smattering of cliches detract from an otherwise very informative review.

Author Response

Thank you for reviewing our manuscript. Please see our attached responses.
